# Predictors of Expectations for the Future Among Young Korean Adults

**DOI:** 10.3390/bs15030391

**Published:** 2025-03-19

**Authors:** Jae-Sun An, Kyung-Hyun Suh

**Affiliations:** Department of Counseling Psychology, Sahmyook University, Seoul 01795, Republic of Korea; lucky603@syu.ac.kr

**Keywords:** stress, gratitude, hardiness ∙ commitment, empathy, expectations for the future

## Abstract

This study explored psychosocial factors related to young adults’ expectations for the future and verified a model that can predict these expectations using psychosocial factors and demographic profiles to provide useful information for further studies and interventions. The participants were 371 Korean adults aged 20–39 years. The predictive models were examined using stepwise regression and decision tree analyses. The results revealed that stress, depression, gratitude, hardiness, interpersonal competence, and social support were significantly correlated with expectations for the future among young adults. Stepwise regression analysis revealed that commitment, reflecting a sense of purpose, and engagement in life accounted for the greatest variance in expectations for the future. Commitment, gratitude, self-directedness, depression, and the presence of disease accounted for approximately 66.7% of the variance in expectations for the future in young adulthood. The decision tree analysis identified commitment as the most important predictor, followed by gratitude, stress, self-directedness, empathy, perceived health, and marital status, showing how these factors are associated with shaping future expectations. These findings suggest that psychological variables such as commitment and gratitude may play a more important role in young adults’ expectations of their future than health or marital status.

## 1. Introduction

Although Korea’s economy has developed rapidly and people’s quality of life has improved, young Korean adults are not only dissatisfied with their lives due to employment and housing difficulties but also have low expectations for their future, often postponing or giving up on marriage and childbirth ([25]; [44]; [43]). In general, young adults tend to have higher expectations and optimism about their future compared to middle-aged and older adults, even in the face of social and economic crises ([21]). However, in the Korean context, young adults’ outlook on their future appears particularly bleak.

In young adulthood, when the future is uncertain, expectations for the future refer to an individual’s anticipation of future life satisfaction, success, and well-being ([11]). This concept reflects one’s belief about whether their future will improve, remain stable, or deteriorate, influencing their motivation, mental health, and decision-making ([23]). Young people’s expectations about the future vary based on personal experiences and social environments and may have been shaped since childhood ([29]). Therefore, this study explores the psychological factors that predict expectations for the future in young adulthood. In other words, we examine the protective and risk factors associated with expectations for the future in this developmental stage.

First, this study assumed that stress is associated with expectations for the future among young adults. Stress is commonly defined as a psychobiological response to external or internal demands that exceed an individual’s perceived ability to cope ([32]; [37]). While stress can be categorized as eustress (positive, motivating stress) or distress (negative, overwhelming stress), this study focuses on distress, as excessive or chronic stress is widely recognized as a factor that negatively impacts mental health and well-being ([51]). However, stress is not inherently harmful; moderate stress can enhance adaptation and resilience ([38]), and eliminating all stress is neither realistic nor beneficial. Mental health and well-being are multifaceted concepts that include hedonic well-being, which refers to pleasure and life satisfaction, as well as eudaimonic well-being, which encompasses personal growth and a sense of meaning ([22]; [47]). In this study, we define mental health and well-being as subjective well-being, resilience, and emotional regulation, which together influence psychological stability and coping ability ([48]; [52]).

Excessive stress is linked to lower life satisfaction and diminished well-being ([61]). Chronic stress can shape negative expectations for the future, reducing confidence in future life satisfaction and success ([42]). Similarly, a recent study found that perceived stress, which refers to an individual’s subjective evaluation of stress levels, was negatively correlated with positive expectations for the future and positively correlated with negative expectations ([42]). Thus, this study examines how perceived stress influences young adults’ future expectations, with a focus on distress. This approach clarifies that higher stress levels are expected to contribute to negative or uncertain expectations rather than positive future anticipation.

Depression in young adulthood may be related to expectations for the future, specifically one’s overall expectations about their future life satisfaction, success, and well-being. Aaron Beck’s cognitive triad theory states that, from a cognitive perspective, depressed individuals tend to have a negative view of themselves, the world, and the future, which can lead to pessimistic expectations about their future prospects ([7]). That is, depression can lower individuals’ satisfaction with their current lives as well as their overall expectations for the future. Empirical studies support this association. For example, [33] ([33]) found a significant negative correlation between depression and expectations for the future. More recently, [35] ([35]) examined the relationship between depression and future expectations among university students, showing that higher levels of depression were linked to lower expectations for the future in general, encompassing aspects of well-being, life satisfaction, and long-term life prospects. This study examines how depression influences young adults’ overall expectations for their future, rather than focusing on specific domains such as career or relationships.

Gratitude is commonly defined as an appreciation for positive experiences and benefits in life, while dispositional gratitude refers to a stable tendency to feel grateful across various situations rather than just in response to specific events ([36]). Research has shown that gratitude is positively linked to subjective well-being and life satisfaction ([4]). Additionally, dispositional gratitude is associated with a positive time perspective, meaning individuals with higher gratitude tend to reflect on their past positively, appreciate the present, and hold optimistic expectations for the future ([45]). Gratitude has also been linked to positive future consequences, referring to the anticipation of beneficial long-term outcomes, such as achieving personal goals and maintaining well-being ([56]). Based on these findings, this study assumes that a stronger tendency toward gratitude will be positively associated with expectations for the future in young adulthood, fostering an optimistic outlook on life.

Psychological hardiness may be related to expectations for the future in young adulthood, especially in times of economic and social uncertainty. Hardiness, introduced by [27] ([27]), is a personality trait characterized by commitment, control, and challenge, enabling individuals to perceive stressors as opportunities for growth and maintain a stable outlook on their future ([54]). Research supports the role of hardiness in fostering positive future expectations. [6] ([6]) found that hardiness protects individuals from stress and strengthens resilience in facing life challenges. Hardiness is also positively linked to life satisfaction and optimism about the future ([42]). Additionally, [12] ([12]) found that hardiness helps individuals reframe stressors, promoting goal-setting and perseverance. Given the economic instability and housing crisis young Korean adults face, hardiness may help them sustain a positive outlook on their future well-being. Thus, this study assumes that higher psychological hardiness will be associated with more positive expectations for the future, helping young adults maintain optimism despite external challenges.

Interpersonal competence refers to an individual’s ability to effectively initiate, maintain, and navigate social interactions across friendships, family, colleagues, and romantic relationships ([5]). It includes communication skills, empathy, and conflict management ([16]). Research suggests that strong interpersonal competence enhances life satisfaction by enabling individuals to form meaningful relationships ([14]). People with higher interpersonal competence tend to have greater confidence in their future relationships, leading to higher expectations for future relationship satisfaction and, consequently, a more positive overall outlook on their future ([5]). Studies have also found that interpersonal competence is positively correlated with future expectations ([1]; [41]). Since interpersonal competence is a broad construct, this study focuses on its core aspects, such as relationship-building ability and empathy rather than all subcomponents. Therefore, this study assumes that young adults with higher interpersonal competence will have more positive expectations for their future.

Social support refers to the perception or experience of being cared for and assisted by others, encompassing emotional, instrumental, informational, and appraisal support ([19]; [3]). It enhances life satisfaction, emotional resilience, and stress reduction, particularly in difficult situations ([15]). Perceived social support has been shown to increase resilience and lower depression, contributing to higher expectations for future well-being ([60]). Research indicates that young adults who receive more social support tend to have greater confidence in achieving personal goals and maintaining stability, leading to higher expectations for their future ([13]; [24]). Social support is particularly linked to well-being and resilience because it provides emotional security and coping resources that help individuals manage stress and maintain a positive outlook on the future ([57]). While social support influences various aspects of future expectations, its effects are strongest in these domains because emotional and instrumental support fosters adaptive coping and psychological stability. Based on these findings, we hypothesize that greater social support will be associated with more positive expectations for the future, particularly in relation to well-being.

This study aimed to investigate psychosocial factors that can predict young Korean adults’ expectations for their future, which can determine their quality of life. To achieve the purpose of this study, the following research questions were examined: First, are there significant relationships between stress, depression, gratitude, hardiness, interpersonal competence, social support, and expectations for the future among young Korean adults? Second, what is the appropriate stepwise regression model for predicting expectations about the future among young Korean adults? Third, what is the appropriate decision tree model for predicting the expectations for the future among young Korean adults?

## 2. Materials and Methods

### 2.1. Research Design and Hypotheses

Based on previous literature and theoretical perspectives, we formulated the following hypotheses:

**H1.** 
*Stress and depression will be negatively associated with expectations for the future in young adults.*


**H2.** 
*Gratitude, hardiness, and interpersonal competence will be positively associated with expectations for the future in young adults.*


**H3.** 
*Social support from family, friends, or superiors will be positively associated with expectations for the future. in young adults.*


**H4.** 
*Age, marital status, perceived health, presence of illness, stress, depression, gratitude, hardiness, interpersonal competence, and social support will significantly contribute to predicting expectations for the future in young adults.*


This study employed a quantitative, correlational, and predictive research design, using survey-based data collection to examine these hypotheses through correlation analysis, regression modeling, and decision tree analysis.

### 2.2. Participants

A total of 371 young Korean adults participated in this study. Their ages ranged from 20 to 39 years, and the mean age was 30.10 ± 5.08 years. The age range for young adulthood in this study was set at 20 to 39 years, based on the developmental psychology literature indicating that young adulthood commonly extends into the late 30s ([31]). Specific information on the characteristics of the participants is presented in the results section.

### 2.3. Participants’ Characteristics

Of the participants, 181 (48.8%) were men and 190 (51.2%) were women (Table 1). Of the participants, 187 were in their 20s (50.4%), and 190 were in their 30s (49.6%). A total of 96 (25.9%) participants had a religious affiliation and 275 (74.1%) reported not having a religion. Additionally, 63 (17.0%) were high school graduates, 278 (74.9%) were college graduates, and 30 (8.1%) had completed or graduated from graduate school.

Among the participants, 92 (26.1%) were married and had a spouse and 274 (73.9%) had a child or children. Additionally, 89 (24.0%) were living alone, and 266 (71.7%) reported having a job. At the time of the survey, 50 participants (13.5%) had a disease.

### 2.4. Data Collection

The data were collected by commissioning Embrain, an online survey company. Institutional review board approval was obtained prior to data collection, and every effort was made to be as ethical as possible during the data collection process. For example, all data for this study were collected with written consent from the participants. After agreeing to participate in the online survey, respondents were informed that they were free to withdraw at any time if they felt uncomfortable while answering the survey. Additionally, they were informed that all data would be used for research purposes only and would be stored on an encrypted computer for three years and then destroyed.

### 2.5. Instruments

#### 2.5.1. Stress (Perceived Stress Scale, PSS-10)

Stress levels experienced by young adults were assessed using the Perceived Stress Scale (PSS-10), developed by [8] ([8]) and adapted into Korean by [34] ([34]). This scale measures the extent to which individuals perceive situations in their lives as stressful over the past month. It consists of 10 items, 5 of which are reverse-scored, rated on a 5-point Likert scale ranging from 1 (never) to 5 (very often), with higher scores indicating greater perceived stress. In the Korean validation study by [34] ([34]), the internal consistency (Cronbach’s α) was 0.82, and the test–retest reliability was 0.66. In this study, Cronbach’s α was 0.81, demonstrating good reliability.

#### 2.5.2. Depression (SU Mental Health Test Subscale)

Depressive symptoms in young adulthood were measuring using the depression subscale of the SU Mental Health Test, developed by [55] ([55]). This subscale is part of a broader 106-item measure designed to assess emotional disorders, personality disorders, and mental health risk and protective factors. For this study, we used the 10-item depression subscale, which has a single-factor structure and was found to be psychometrically stable in relation to anxiety and hypomania. Example items include “I feel no interest in life” and “I often feel like I am the only one in the world”. The items were rated on a 6-point Likert scale ranging from 1 (strongly disagree) to 6 (strongly agree), with higher scores indicating higher levels of depressive symptoms. The internal consistency of the depression subscale was 0.91 in the original scale development study ([55]), with a test–retest reliability of 0.79. In this study, Cronbach’s α was 0.95, demonstrating excellent reliability.

#### 2.5.3. Gratitude (Gratitude Questionnaire-6, GQ-6)

The Gratitude Questionnaire-6 (GQ-6), developed by [36] ([36]) and validated in Korean by [30] ([30]), was used to measure young adults’ general disposition toward gratitude. This scale assesses an individual’s tendency to recognize, appreciate, and express gratitude in various aspects of life rather than targeting gratitude toward specific individuals. Example items include “In my life, I have a lot to be thankful for” and “When I look at the world, I don’t see much to be grateful for” (reverse-scored). The GQ-6 consists of six items with a single-factor structure, including two reverse-scored items, and is rated on a 7-point Likert scale ranging from 1 (strongly disagree) to 7 (strongly agree). The internal consistency (Cronbach’s α) of the scale was 0.85 in a validation study ([30]), and 0.91 in this study.

#### 2.5.4. Hardiness (Brief Measure of Hardiness, BMH)

Hardiness in young adults was measured using the Brief Measure of Hardiness (BMH), developed by [53] ([53]). This scale assesses an individual’s psychological hardiness through three subscales: commitment, self-directedness, and tenacity, each comprising four items. Example items include “I live a life full of interest” (commitment), “My life is determined by my own decisions” (self-directedness), and “I believe that hardships make me stronger” (tenacity). It is rated on a 6-point Likert scale ranging from 1 (not at all true) to 6 (very true), with higher scores indicating stronger hardiness. The scale demonstrated a stable factorial structure and good criterion validity in its development study ([53]), where internal consistency (Cronbach’s α) was 0.91 for commitment, 0.85 for self-directedness, 0.89 for tenacity, and 0.88 for all 12 items, with a test–retest reliability of 0.77. In this study, Cronbach’s α was 0.88 for commitment, 0.85 for self-directedness, 0.86 for tenacity, and 0.91 for all items.

#### 2.5.5. Interpersonal Competence (SU Mental Health Test Subscale)

Interpersonal competence in early adulthood was measured using the interpersonal competence subscale from the SU Mental Health Test, developed by [55] ([55]). This subscale evaluates the protective factors of mental health and consists of two components: the ability to form interpersonal relationships (3 items) and empathy (3 items). Example items include “I like meeting new people” (ability to form interpersonal relationships) and “I empathize with others’ feelings” (empathy). Each item is rated on a 6-point Likert scale ranging from 1 (strongly disagree) to 6 (strongly agree), with higher scores indicating stronger interpersonal competence. The scale development study demonstrated a stable factorial structure and satisfactory reliability, with internal consistency (Cronbach’s α) of 0.83 for the ability to form interpersonal relationships, 0.75 for empathy, and 0.80 for all items, and test–retest reliability of 0.75, 0.74, and 0.78, respectively ([55]). In this study, Cronbach’s α was 0.85 for the ability to form interpersonal relationships, 0.81 for empathy, and 0.86 for all items, confirming satisfactory reliability.

#### 2.5.6. Social Support (Social Support Scale)

Social support perceived by young adults was measured using the Social Support Scale, developed by [18] ([18]) and translated and validated in Korean by [40] ([40]). This scale assesses whether individuals perceive themselves as receiving support from different sources, including superiors (seniors), colleagues (friends), and family, in the form of problem-solving assistance, dependability, active listening, and emotional support. In this study, support from friends, seniors, superiors, and colleagues was measured as a combined category rather than separately, along with a distinct measure of family support. Each category was assessed with four items, and the total score for each of these two sources (friends/superiors/colleagues and family) was used in the analysis rather than separate subscale scores. Example items include “My friends (or seniors) help me solve problems” (friends/superiors/colleagues support) and “My family provides emotional support when I share personal concerns” (family support). Responses were rated on a 5-point Likert scale ranging from 1 (never) to 5 (very often), with higher scores indicating greater perceived social support. The internal consistency (Cronbach’s α) was 0.89 in [40]’s ([40]) validation study and 0.91 in this study, demonstrating strong reliability.

#### 2.5.7. Expectations for the Future (Life Satisfaction Expectancy Scale)

Expectations for the future in young adulthood were measured using the Life Satisfaction Expectancy Scale, developed by [23] ([23]). This scale was adapted from the Satisfaction with Life Scale ([9]), modifying statements about current life satisfaction into future-oriented expectations. For instance, the original item “The conditions of my life are excellent” was modified to “In the future, the conditions of my life will become more excellent”. Other example items include “In the future, I will be more satisfied with my life” and “Going forward, I will develop and grow further.” The scale consists of five items, each rated on a 7-point Likert scale ranging from 1 (strongly disagree) to 7 (strongly agree), with higher scores indicating more positive expectations for the future. The internal consistency (Cronbach’s α) of the scale was 0.96 in this study, demonstrating excellent reliability.

### 2.6. Statistical Analysis

The Statistical Package for the Social Sciences for Windows (version 25.0) was used for all data analyses. Prior to the analysis, we checked the skewness and kurtosis of the variables to confirm their normal distributions. Correlation and stepwise regression analyses were performed as parametric statistics, and decision tree analysis was performed as non-parametric statistics.

In this study, the chi-square automatic interaction detection (CHAID) method was used in decision tree analysis. This method was introduced by [20] ([20]), an algorithm that performs multiple separations based on the *χ*^2^ value from cross-tabulations and the F value from analysis of variance. The total score of expectations for the future was selected as the target variable, and the likelihood ratio *χ*^2^ value was used because it is a continuous variable. The maximum number of levels was 3, and the minimum number of cases for the parent and child nodes was set to 40 and 10, respectively.

## 3. Results

### 3.1. Relationships Between the Variables Involved in This Study

Table 2 shows the results of the correlation analysis for stress, depression, gratitude, hardiness, interpersonal competence, received social support, and expectations for the future among young Korean adults (Table 2). None of the absolute values of skewness and kurtosis exceeded 1.0, indicating that the variances of all variables did not significantly deviate from the normal distribution, satisfying the conditions for parametric statistics ([59]).

The correlation analysis revealed that stress (*r* = −0.56, *p* < 0.001) and depression (*r* = −0.66, *p* < 0.001) were negatively correlated with expectations for the future, whereas gratitude (*r* = 0.69, *p* < 0.001), hardiness (*r* = 0.75, *p* < 0.001), interpersonal competence (*r* = 0.58, *p* < 0.001), and received social support (*r* = 0.59, *p* < 0.001) were positively correlated with.

Stress was positively correlated with depression in young adulthood (*r* = 0.69, *p* < 0.001), whereas it was negatively correlated with young adults’ gratitude (*r* = −0.47, *p* < 0.001), hardiness (*r* = −0.51, *p* < 0.001), interpersonal competence (*r* = −0.35, *p* < 0.001), and social support (*r* = −0.44, *p* < 0.001). Depression in young adulthood was negatively correlated with young adults’ gratitude (*r* = −0.66, *p* < 0.001), hardiness (*r* = −0.61, *p* < 0.001), interpersonal competence (*r* = −0.42, *p* < 0.001), and social support (*r* = −0.59, *p* < 0.001). Moreover, positive associations were found between gratitude, hardiness, interpersonal competence, and received social support.

### 3.2. Predictive Models for Expectations for the Future

We verified models predicting expectations for the future among young adults. First, we conducted a stepwise regression analysis with demographic profiles, variables related to health conditions, and psychological variables or their sub-factors, including a correlation analysis. Multicollinearity among all the variables was checked. If the tolerance is less than 0.2 or 0.1 and the variance inflation factor (VIF) is higher than 5.0 or 10.0, multicollinearity problems may occur ([26]). The tolerance of the predictors was 0.359–0.526, the VIF was 1.900–2.786, and multicollinearity was not significant.

Table 3 shows that commitment was the determinant predictor of the expectations for the future (*β* = 0.732, *p* < 0.001), followed by gratitude (*β* = 0.372, *p* < 0.001), self-directedness (*β* = 0.187, *p* < 0.001), depression (*β* = −0.197, *p* < 0.001), and having a disease (*β* = −0.073, *p* < 0.05) in this stepwise regression model. Commitment accounted for approximately 53.5% of the variance in expectations for the future in young adulthood. In addition to commitment, gratitude accounted for approximately 8.2% of the variance in expectations for the future. Self-directedness in the young adulthood accounted for approximately an additional 2.5% of the variance in expectations for the future. Depression accounted for approximately an additional 2.0% of the variance. In addition, the presence of disease accounted for approximately 0.5% of the variance in the model. The five predictors included in this model accounted for approximately 66.7% of the variance in expectations for the future in young adulthood.

To verify a decision tree model that could predict young adults’ expectations for their future, psychological variables or sub-factors of variables and demographic profiles, including parametric and non-parametric data, and variables related to health conditions were entered as potential predictors.

The results revealed that the total number of nodes was 23, the number of terminal nodes was 15, and the number of depths was 3. The risk estimate was 16.36 (*SE* = 1.59), and the average risk estimate of the 10-fold cross-validation was 21.27 (*SE* = 1.91), indicating differences within the margin of error. The average of the root node’s expectations for the future was 23.18, and 11 nodes (Nodes 3, 4, 5, 13, 15, 16, 18, 19, 20, 21, and 22) exceeded the average; young adults belonging to them showed higher scores on expectations for the future (Figure 1). Table 4 shows that the order of gain for nodes was 5 (8.4%), 16 (8.4%), 19 (3.8%), 22 (8.1%), 20 (7.3%), 21 (3.8%), 18 (11.3%), 14 (3.2%), 10 (3.8%), 17 (15.1%), 6 (3.0%), 11 (8.9%), 8 (5.1%), 7 (4.3%), and 9 (5.7%).

In this model, the first criterion used to classify the level of expectations for the future in young adulthood was commitment (see Figure 1).

The average expectation for the future of 67 participants with a commitment score of 9 or less was very low at 14.91 (Node 1). The average expectation for the future of the 47 participants (commitment scores were over 9 and 11 or less), the 139 participants (commitment scores were between 11 and 15), the 87 participants (commitment scores were between 15 and 19) was 19.19, 23.65, and 27.53, respectively; the average expectation for the future of young adults with commitment scores over 19 was the highest at 32.77 (Node 5). There were 31 participants with a commitment score of between 15 and 19 and a gratitude score of over 35 who had a higher average for expectations for their future at 30.23 (Node 16).

Additionally, 44 participants with a commitment score of between 15 and 19 and a gratitude score between 26 and 35 had a relatively higher average expectation for the future at 26.91. Among them, young adults with a self-directedness score of over 17 had a higher average of expectations for the future at 28.03 (Node 22).

When the commitment score was between 11 and 15, and the empathy score was over 13, the average expectations for the future was 27.10 (Node 13). Among them, married participants had a higher average expectation of the future at 30.07 (node 19). However, when the commitment score was between 11 and 15 and the empathy score was over 13, the average of expectations for the future was at 22.20; young adults with a self-directedness score of over 16 had a higher average for expectations for the future at 24.05 (Node 18).

Even if the commitment score was between 9 and 11, if the young adults perceived themselves as healthy, the average of their expectations for their future was 22.50 (Node 10). If the commitment score was 9 or less and the stress score was over 37, the average expectation for the future was the lowest at 10.76 (Node 9).

## 4. Discussion

This study explored the variables that could predict expectations for the future among young Korean adults to provide useful information for future studies and interventions for well-being in young adulthood. Analyses show the relationships between psychosocial factors, such as stress, depression, gratitude, hardiness, interpersonal competence, and social support, which are considered to be associated with expectations for the future in young adulthood. Additionally, demographic profiles, including categorical data that may be related to expectations for the future, were entered into the predictive model. In particular, a decision tree analysis, a data mining method, was conducted, and significant results were derived. The implications of the findings are as follows.

In this study, the more stress young adults perceived themselves to experience, the lower their expectations for their future, with perceived stress accounting for 31.6% of the variance in future expectations. This finding aligns with previous research indicating that stress not only reduces life satisfaction ([61]) but can also significantly diminish individuals’ confidence and optimism about their future prospects. While we did not measure current life satisfaction directly, our results suggest that high levels of stress may interfere with future-oriented thinking, contributing to a pessimistic outlook

Additionally, the decision tree model revealed that the impact of perceived stress on future expectations was greater when individuals had lower levels of commitment. Commitment, a component of psychological hardiness, reflects a person’s engagement with life, curiosity, and investment in meaningful activities. This interaction suggests that young adults with low commitment may be particularly vulnerable to stress, as they may lack the psychological resources needed to reframe stressful experiences or maintain a hopeful outlook. Conversely, those with higher commitment may be more resilient in maintaining positive expectations despite experiencing stress.

Depression in young adulthood is closely related to expectations for the future, as individuals experiencing depression often struggle with pessimistic thoughts about their long-term prospects ([35]). In this study, depression accounted for approximately 43.3% of the variance in future expectations, indicating a substantial impact. The more depressed young adults were, the lower their expectations for future well-being, career satisfaction, and personal fulfillment. This aligns with recent findings that depressive symptoms negatively influence individuals’ future orientation, goal-setting, and perceived self-efficacy ([17]; [62]).

Additionally, depression was identified as a significant predictor in the stepwise regression model, which is consistent with [7]’s ([7]) “cognitive triad” theory, suggesting that depression leads individuals to hold negative views of themselves, the world, and their future. Studies also indicate that depressed individuals are more likely to experience biased or overly pessimistic future expectations, which affects their motivation and decision-making processes ([62]).

Given these findings, addressing depression is essential for promoting positive future expectations in young adults. Clinical interventions such as cognitive–behavioral therapy (CBT) and future-directed therapy (FDT) have been shown to improve future outlook by restructuring negative thought patterns ([58]). Additionally, broader preventive measures, including social support systems and mental health education, may help mitigate the long-term impact of depressive symptoms on future expectations ([39]).

This study found that young adults with high-gratitude tendencies had higher expectations of their futures. The effect size of gratitude on expectations for the future in young adults was approximately 0.473. Gratitude was also a significant predictor in a stepwise regression model that predicted expectations for the future in young adults. In the decision tree model, expectations for the future of young adults with relatively high commitment differed according to their level of gratitude. This may indicate that young adults, who perceive that they have experienced many things to be grateful for, expect that there will be many such things in the future. Gratitude interventions have often been applied to promote mental health and well-being ([28]), but this study suggests that such interventions may also be effective in helping young adults improve their expectations for their future.

In this study, hardiness was positively associated with expectations for the future, with an effect size of 0.557, making it a strong predictor. To better understand its influence, we analyzed its sub-factors separately, identifying commitment as the most influential predictor in both the decision tree and stepwise regression models. Since commitment alone explained over half of the variance, it is clear that it plays a dominant role, while self-directedness, gratitude, and depression made much smaller contributions. In the stepwise regression model, commitment, gratitude, depression, and self-directedness together accounted for 66.7% of the variance, but given that commitment alone explained the majority, it is evident that this factor drives the relationship. This suggests that intervention efforts to enhance young adults’ future expectations should primarily focus on strengthening commitment, rather than broadly targeting other psychological traits. Future research should explore why commitment plays such a dominant role and examine effective strategies for fostering it in young adults.

This study found that the more competency young Korean adults demonstrated in interpersonal relationships, the more positive their expectations for their future. These findings reinforce previous research indicating that interpersonal competence fosters not only satisfaction with present life but also optimism about the future ([1]; [5]; [41]). In particular, empathy—a sub-factor of interpersonal competence—was included in the decision tree model predicting young adults’ expectations for their future. Specifically, young adults with medium levels of commitment had lower-than-average expectations if they had poor empathy but higher-than-average expectations if they demonstrated strong empathy. This suggests that empathy plays a role in shaping future expectations, particularly when other protective psychological factors, such as commitment, are moderate rather than high.

While previous research has shown that empathy is positively correlated with young people’s life satisfaction ([10]), this study suggests that young adults who are more empathetic also tend to expect their lives to be satisfactory in the future. Given that empathy may influence how individuals perceive and anticipate social interactions and support from others ([2]), it is possible that higher empathy fosters more optimistic future expectations by shaping positive social experiences and support networks. Although interpersonal skills training, including empathy training, has been shown to improve social functioning and well-being, its direct impact on future expectations remains an open question. Future research should examine whether and how interpersonal competence training could influence young adults’ expectations for the future and in what specific domains (e.g., career, relationships, and personal well-being) this effect might be observed.

As hypothesized, the more social support young adults receive, the more positive expectations they have for their future, particularly in terms of personal well-being. This study reinforces previous findings that social support enhances quality of life in young adulthood and can also contribute to a more optimistic future outlook ([13]; [24]). In the decision tree model, among young adults with high empathy, those who were married had higher expectations for their future compared to unmarried individuals. However, this result does not imply that marriage alone predicts positive future expectations. It is possible that married young adults in this study benefited from emotional support within their relationships, which may have contributed to their more optimistic future outlook. However, we did not measure relationship quality, spousal support, or marital satisfaction, so further research is needed to examine whether perceived support within a marriage, rather than marital status itself, influences future expectations.

Additionally, while empathy was included in the model, we cannot conclude that higher empathy among married individuals directly explains their increased future expectations. Empathy, as measured in this study, refers to the ability to understand and share others’ emotions, but it does not necessarily indicate how individuals provide emotional support in relationships. Future studies should explore whether higher empathy contributes to stronger relationship satisfaction and whether this, in turn, influences expectations for the future. Given these findings, future research should investigate the role of relationship quality, perceived emotional support, and interpersonal dynamics in shaping young adults’ expectations for their future, rather than assuming a direct link between marriage and positive future outlooks.

In this study, young adults who had been diagnosed with a physical illness or perceived themselves as unhealthy had lower expectations for their future. Additionally, perceived health status was included as a significant predictor in the decision tree model, highlighting its role in shaping young adults’ expectations about their well-being and life prospects. However, it is important to interpret these findings cautiously, as this study did not differentiate between types of illnesses, which can range from minor, temporary conditions to chronic or life-threatening diseases. Previous research suggests that how individuals appraise their illness, rather than the illness itself, plays a critical role in its psychological impact ([49]). Since this study did not explicitly measure illness appraisal, further research is needed to determine whether subjective perceptions of illness severity mediate the relationship between health status and future expectations.

Moreover, the assertion that interventions could help young adults with physical illnesses develop a more positive attitude toward their future requires further empirical support. Psychological interventions for young people with chronic or severe health conditions have been shown to be beneficial, particularly expectation-focused interventions that help individuals reframe their future outlook and adjust their coping mechanisms ([46]). Additionally, supportive therapy and social interventions aimed at fostering resilience and adaptive coping strategies have been effective in improving the mental health and future orientation of young adults with chronic illnesses ([50]). Future research should explore which specific types of interventions are most effective in enhancing future expectations among young adults with different health conditions.

## 5. Limitations of the Study

The current study had several limitations. First, while the sample included young adults from various regions of Korea, it was drawn from an online survey panel, which may limit its generalizability to all young Korean adults. However, since participants were recruited from a nationally representative panel, the sample still provides meaningful insights.

Second, although we discuss possible causal relationships between variables in some parts of the discussion, this study was correlational in nature. Therefore, causal inferences cannot be made, and future research should consider longitudinal or experimental designs to better examines causal relationships between psychosocial factors and future expectations.

Third, this study examined interpersonal competence as a predictor of future expectations. Due to its broad nature, we focused on relationship-building ability and empathy, rather than assessing all subcomponents. Future research should consider a more comprehensive approach to fully capture how interpersonal competence influences future expectations.

Fourth, we did not analyze whether the relationships between psychosocial factors and future expectations varied between young adults in their 20s and 30s. Given that young adulthood encompasses different stages of life, future research should explore potential age-related differences to better understand how factors such as marital status, employment, and financial stability influence future expectations.

Fifth, while decision tree analysis has the advantage of being hypothesis-free and accommodating both numerical and categorical predictors, its application in SPSS is somewhat limited compared to more sophisticated machine learning approaches. Future research should explore alternative methodologies such as ensemble decision tree models or deep learning approaches for more reliable predictive modeling.

## 6. Conclusions

Commitment, gratitude, self-directedness, depression, stress, empathy, perceived health, having a disease, and the presence of a spouse were also predictors of expectations for the future in young adulthood. These findings suggest that to help young adults develop more positive expectations for their future, mental health professionals should implement interventions that strengthen their sense of commitment, encourage gratitude, and enhance self-directedness. Although this study has certain limitations, it contributes to the growing body of research on young adults’ attitudes toward their future and provides practical insights for interventions aimed at fostering positive future expectations among young Korean adults. This study found that perceived stress, depression, gratitude, hardiness, interpersonal competence, and social support were significantly associated with expectations for the future in young adulthood. Among these, commitment emerged as the strongest predictor, emphasizing its crucial role in shaping future expectations. Other significant predictors included gratitude, self-directedness, depression, stress, empathy, perceived health, physical illness, and the presence of a spouse. These findings suggest that mental health professionals should prioritize fostering commitment, gratitude, and self-directedness in young adults to help promote a more optimistic outlook on their future. Furthermore, interventions focusing on resilience-building, interpersonal skill development, and stress management could further support young adults in navigating uncertainty and enhancing their expectations for the future. Future research should continue to investigate the long-term effects of these psychological factors on future expectations and explore interventions that can effectively enhance optimism and well-being in young adulthood.

## Figures and Tables

**Figure 1 behavsci-15-00391-f001:**
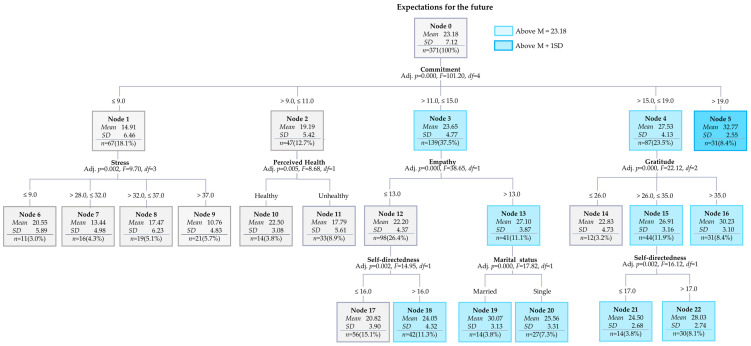
Decision tree model of expectations for the future among young Korean adults.

**Table 1 behavsci-15-00391-t001:** Characteristics of participants (*N* = 371).

Variables	Frequency	Percent (%)
Gender		
Man	181	48.8
Woman	190	51.2
Age		
20s	187	50.4
30s	184	49.6
Religion		
Having religion	96	25.9
None	275	74.1
Educational attainment		
High school	63	17.0
College	278	74.9
Graduate school	30	8.1
Marital status		
Married	92	26.2
Single	272	73.3
Divorced	2	0.5
Presence of children		
Having a child	58	15.6
None	313	84.4
Housing type		
Living alone	89	24.0
Living with other(s)	282	76.0
Occupation		
Having a job	266	71.7
None	105	28.3
Presence of disease		
Having a disease	50	13.5
None	321	86.5

**Table 2 behavsci-15-00391-t002:** Correlational matrix of stress, depression, gratitude, hardiness, interpersonal competence, social support, and expectations for the future.

Variables	1	2	3	4	5	6	7
1. Stress	1						
2. Depression	0.69 ***	1					
3. Gratitude	−0.47 ***	−0.66 ***	1				
4. Hardiness	−0.51 ***	−0.61 ***	0.69 ***	1			
5. Interpersonal competence	−0.35 ***	−0.42 ***	0.57 ***	0.65 ***	1		
6. Social support	−0.44 ***	−0.59 ***	0.70 ***	0.63 ***	0.58 ***	1	
7. Expectations for the future	−0.56 ***	−0.66 ***	0.69 ***	0.75 ***	0.58 ***	0.59 ***	1
*M*	29.87	27.81	29.04	47.53	23.39	27.85	23.18
*SD*	5.62	12.01	6.94	9.67	5.36	6.15	7.12
Skewness	0.13	0.36	−0.06	0.04	−0.01	−0.12	−0.48
Kurtosis	0.44	−0.66	−0.37	−0.03	−0.05	−0.25	−0.20

*** *p* < 0.001. Pearson correlation coefficients.

**Table 3 behavsci-15-00391-t003:** Results of the stepwise regression analysis of expectations for the future.

Variables	*B*	*β*	*t*	∆*R*^2^	*F*
Commitment	1.190	0.732	20.61 ***	0.535	146.04 ***
Gratitude	0.382	0.372	8.87 ***	0.082	
Self-directedness	0.374	0.187	5.10 ***	0.025	
Depression	−0.117	−0.197	−4.57 ***	0.020	
Having a disease	1.531	−0.073	−2.37 *	0.005	

* *p* < 0.05, *** *p* < 0.001.

**Table 4 behavsci-15-00391-t004:** Gain summary for nodes.

Nodes	*N*	%	*M*
5	31	8.4	32.77
16	31	8.4	30.23
19	14	3.8	30.07
22	30	8.1	28.03
20	27	7.3	25.56
21	14	3.8	24.50
18	42	11.3	24.05
14	12	3.2	22.83
10	14	3.8	22.50
17	56	15.1	20.82
6	11	3.0	20.55
11	33	8.9	17.79
8	19	5.1	17.47
7	16	4.3	13.44
9	21	5.7	10.76

Growing method: CHAID.

## Data Availability

The datasets used and analyzed in this study can be obtained from the corresponding author upon reasonable request.

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
