# Peer review of "Predictors of Expectations for the Future Among Young Korean Adults"

_behavsci, 2025, doi:10.3390/bs15030391_

Round 1
Reviewer 1 Report
Comments and Suggestions for Authors
1. In Table one put please a title for the second column.
2. Please indicate the software, modules if any or procedures used to perform all analyses.
3. Please indicate in the section of the instruments more details about what each instrument measures. For example, at gratitude, what does the instrument measure? Gratitude towards whom? There are more options.
4. For Hardness the same. What the instrument measures? The title is not sufficient.
5. Provide an example of items for each measure described too. It clarifies what you refer to.
6. For social support, did you use the total scale or the individual scales. Specify. It should be clear even from this point.
7. Describe the instrument expectations for the future in more detail. It is an instrument that measures expectations in general or specific refering to something? Reading the article it is ambiguous what you refer to.
8. Write that it is Pearson correlations for the correlation Matrix.
9. Provide the unstandardized regression coeficients too in the regression too.
10. Specify or explain this paragraph:
"When the commitment score was between 11 and 15, and the empathy score was over 13, the average of expectations for the future was 27.10 (Node 13). Among them, married participants had a higher average expectation of the future at 30.07 (node 19). However, when the commitment score was between 11 and 15 and the empathy score was over 13, the average of expectations for the future was at 22.20; young adults with a self-directedness score of over 16 had a higher average for expectations for the future at 24.05 (Node 18). Even if the commitment score was between 9 and 11, if the young adults perceived themselves as healthy, the average of their expectations for their future was 22.50 (Node 10). If the commitment score was 9 or less and the stress score was over 37, the average expectation for the future was the lowest at 10.76 (Node 9)."
Why did you present some of the routes and not some others? Explain that those routes are the most probable or evaluate them somehow. Why did you choose to explain these and not the others?
11. Present the hypotheses clearly. You mention them in discussion section. But they do not appear in the Methodology section.
Author Response
Thank you very much for taking the time to review this manuscript. Please find the detailed responses below and the corresponding revisions/corrections highlighted/in track changes in the re-submitted files. The revised parts were marked in red, and we included the page and line of the revised part. We appreciate your complimentary comments. We have omitted our response to your kind words here.
Point-by-point response to Comments and Suggestions for Authors
Comments 1: In Table one put please a title for the second column.
Response 1: Thank you for your valuable comments. We have incorporated your suggestion by adding the "Frequency" title to the second column of Table 1. The revised table now clearly displays column headings, improving readability and clarity. [Line 175]
Comments 2: Please indicate the software, modules if any or procedures used to perform all analyses.
Response 2: Thank you for your feedback. We appreciate your feedback regarding the indication of software, modules, or procedures used for the analyses. In our manuscript, we have provided this information as follows: [Line 284-285]
The Statistical Package for the Social Sciences for Windows (version 25.0) was used for all data analyses.
Comments 3: Please indicate in the section of the instruments more details about what each instrument measures. For example, at gratitude, what does the instrument measure? Gratitude towards whom? There are more options.
Response 3: Thank you for your valuable feedback. n response to your comment, we have clarified that the Gratitude Questionnaire-6 (GQ-6) measures an individual's general disposition toward gratitude, rather than gratitude toward specific individuals such as family, friends, or colleagues. This scale assesses the extent to which individuals recognize, appreciate, and express gratitude in various aspects of life. To enhance clarity, we have also included example items, such as "In my life, I have a lot to be thankful for" and "When I look at the world, I don’t see much to be grateful for" (reverse-scored), to illustrate the nature of the gratitude being measured. We appreciate your valuable suggestions and welcome any further recommendations for improvement. [Line 212-222]
The Gratitude Questionnaire-6 (GQ-6), developed by McCullough et al. (2002) and validated in Korean by Kwon et al. (2006), was used to measure young adults' general disposition toward gratitude. This scale assesses an individual's tendency to recognize, appreciate, and express gratitude in various aspects of life rather than targeting gratitude toward specific individuals. Example items include "In my life, I have a lot to be thankful for" and "When I look at the world, I don’t see much to be grateful for" (reverse-scored). The GQ-6 consists of six items with a single-factor structure, including two reverse-scored items, and is rated on a 7-point Likert scale ranging from 1 (strongly disagree) to 7 (strongly agree). The internal consistency (Cronbach’s α) of the scale was 0.85 in a validation study (Kwon et al., 2006), and 0.91 in this study.
Comments 4: For Hardness the same. What the instrument measures? The title is not sufficient.
Response 4: Thank you for your valuable feedback. In response to your comment, we have clarified that the Brief Measure of Hardiness (BMH) assesses an individual's psychological resilience and adaptability in the face of challenges. Specifically, this scale measures three key components of hardiness: commitment (engagement and sense of purpose in life), self-directedness (belief in personal control over life outcomes), and tenacity (persistence in overcoming difficulties). To provide further clarity, we have also refined the titles of other measurement instruments to better reflect what each scale assesses. Additionally, we have included example items, such as "I live a life full of interest" (commitment), "My life is determined by my own decisions" (self-directedness), and "I believe that hardships make me stronger" (tenacity), to illustrate the constructs being measured. We appreciate your insightful suggestions and welcome any further recommendations for improvement. [Line 224-236]
Based on your advice, we have refined the titles of other measurement instruments in addition to the Hardiness Scale to make them more specific.
Hardiness in young adults was measured using the Brief Measure of Hardiness (BMH), developed by Suh (2022). This scale assesses an individual's psychological haridiness through three subscales: commitment, self-directedness, and tenacity, each comprising four items. Example items include "I live a life full of interest" (commitment), "My life is determined by my own decisions" (self-directedness), and "I believe that hardships make me stronger" (tenacity). It is rated on a 6-point Likert scale ranging from 1 (not at all true) to 6 (very true), with higher scores indicating stronger hardiness. The scale demonstrated a stable factorial structure and good criterion validity in its development study (Suh, 2022), where internal consistency (Cronbach’s α) was 0.91 for commitment, 0.85 for self-directedness, 0.89 for tenacity, and 0.88 for all 12 items, with a test-retest reliability of 0.77. In this study, Cronbach’s α was 0.88 for commitment, 0.85 for self-directedness, 0.86 for tenacity, and 0.91 for all items.
Comments 5: Provide an example of items for each measure described too. It clarifies what you refer to.
Response 5: Thank you for your insightful feedback. In response to your suggestion, we have included example items for each measurement instrument to clarify what each scale assesses. For instance, in the Gratitude Questionnaire-6 (GQ-6), an example item is "In my life, I have a lot to be thankful for", illustrating the general disposition toward gratitude. In the Brief Measure of Hardiness (BMH), items such as "I live a life full of interest" (commitment) and "I believe that hardships make me stronger" (tenacity) highlight key aspects of psychological resilience. Similarly, for Interpersonal Competence, an item like "I like meeting new people" represents the ability to form interpersonal relationships, while "I empathize with others' feelings" reflects empathy. We believe these additions enhance clarity and precision in describing the constructs measured. We appreciate your valuable input and welcome any further suggestions for improvement. [Line 204-277]
Example items include "I feel no interest in life" and "I often feel like I am the only one in the world
Example items include "In my life, I have a lot to be thankful for" and "When I look at the world, I don’t see much to be grateful for" (reverse-scored).
Example items include "I live a life full of interest" (commitment), "My life is determined by my own decisions" (self-directedness), and "I believe that hardships make me stronger" (tenacity)
Example items include "I like meeting new people" (ability to form interpersonal relationships) and "I empathize with others' feelings" (empathy).
Example items include "My friends (or seniors) help me solve problems" (superiors/colleagues support) and "My family provides emotional support when I share personal concerns" (family support).
For instance, the original item "The conditions of my life are excellent" was modified to "In the future, the conditions of my life will become more excellent." Other example items include "In the future, I will be more satisfied with my life" and "Going forward, I will develop and grow further."
Comments 6: For social support, did you use the total scale or the individual scales. Specify. It should be clear even from this point.
Response 6: Thank you for your valuable comment. In response to your comment, we have clarified that in this study, support from friends, seniors, superiors, and colleagues was measured as a single category, rather than separately, along with a distinct measure of family support. Additionally, we have specified that only the total scores for these two sources of support (friends/superiors/colleagues and family) were used in the analysis, rather than individual subscale scores. We have updated the description accordingly to ensure clarity. We appreciate your insightful suggestions and welcome any further recommendations for improvement. [Line 253-268]
Social support perceived by young adults was measured using the Social Support Scale, developed by Iverson et al. (1998) and translated and validated in Korean by No (2011). This scale assesses whether individuals perceive themselves as receiving support from different sources, including superiors (seniors), colleagues (friends), and family, in the form of problem-solving assistance, dependability, active listening, and emotional support. In this study, support from friends, seniors, superiors, and colleagues was measured as a combined category rather than separately, along with a distinct measure of family support. Each category was assessed with four items, and the total score for each of these two sources (friends/superiors/colleagues and family) was used in the analysis rather than separate subscale scores. Example items include "My friends (or seniors) help me solve problems" (friends/superiors/colleagues support) and "My family provides emotional support when I share personal concerns" (family support). Responses were rated on a 5-point Likert scale ranging from 1 (never) to 5 (very often), with higher scores indicating greater perceived social support. The internal consistency (Cronbach’s α) was 0.89 in No’s (2011) validation study and 0.91 in this study, demonstrating strong reliability.
Comments 7: Describe the instrument expectations for the future in more detail. It is an instrument that measures expectations in general or specific refering to something? Reading the article it is ambiguous what you refer to.
Response 7: Thank you for your insightful feedback. In response to your comment, we have clarified that the Life Satisfaction Expectancy Scale (Kim, 2007) measures general expectations for the future, rather than expectations related to a specific domain (e.g., career, relationships, or health). This instrument assesses an individual's overall optimism about future life satisfaction and personal growth, based on modifications of the original Satisfaction with Life Scale (Diener et al., 1985) to a future-oriented perspective. For instance, the original item "The conditions of my life are excellent" was adapted to "In the future, the conditions of my life will become more excellent." Example items include "In the future, I will be more satisfied with my life" and "Going forward, I will develop and grow further." To enhance clarity, we have explicitly stated that this scale measures broad life expectations rather than domain-specific expectations. We appreciate your valuable suggestions and welcome any further recommendations. [Line 270-281]
Expectations for the future in young adulthood were measured using the Life Satisfaction Expectancy Scale, developed by Kim (2007). This scale was adapted from the Satisfaction with Life Scale (Diener et al., 1985), modifying statements about current life satisfaction into future-oriented expectations. For instance, the original item "The conditions of my life are excellent" was modified to "In the future, the conditions of my life will become more excellent." Other example items include "In the future, I will be more satisfied with my life" and "Going forward, I will develop and grow further." The scale consists of five items, each rated on a 7-point Likert scale ranging from 1 (strongly disagree) to 7 (strongly agree), with higher scores indicating more positive expectations for the future. The internal consistency (Cronbach’s α) of the scale was 0.96 in this study, demonstrating excellent reliability.
Comments 8: Write that it is Pearson correlations for the correlation Matrix.
Response 8: Thank you for your advice. We have incorporated your suggestion by specifying that the correlation matrix presents Pearson correlation coefficients for clarity. [Line 311]
Comments 9: Provide the unstandardized regression coeficients too in the regression too.
Response 9: Thank you for your feedback. We have incorporated your suggestion by including the unstandardized regression coefficients (B values) in the table. [Line 341]
Comments 10: Specify or explain this paragraph:
"When the commitment score was between 11 and 15, and the empathy score was over 13, the average of expectations for the future was 27.10 (Node 13). Among them, married participants had a higher average expectation of the future at 30.07 (node 19). However, when the commitment score was between 11 and 15 and the empathy score was over 13, the average of expectations for the future was at 22.20; young adults with a self-directedness score of over 16 had a higher average for expectations for the future at 24.05 (Node 18). Even if the commitment score was between 9 and 11, if the young adults perceived themselves as healthy, the average of their expectations for their future was 22.50 (Node 10). If the commitment score was 9 or less and the stress score was over 37, the average expectation for the future was the lowest at 10.76 (Node 9)."
Why did you present some of the routes and not some others? Explain that those routes are the most probable or evaluate them somehow. Why did you choose to explain these and not the others?
Response 10: Thank you for your valuable feedback. We selected and presented the specific routes in the decision tree based on their statistical significance, interpretability, and relevance to the research objectives. The chosen paths represent the most probable and meaningful variations in expectations for the future, as they highlight the key predictors—commitment, empathy, gratitude, self-directedness, and stress—and their notable interactions.
We focused on the most prominent branches where the variation in expectations for the future was substantial, ensuring a clear and structured narrative. Additionally, the paths we described illustrate meaningful comparisons, such as cases where higher self-directedness or marital status led to greater expectations for the future and how stress and low commitment levels were associated with the lowest scores.
However, we acknowledge that alternative routes could be explored, and we are open to further expanding our explanation if needed. If you believe additional details would strengthen our interpretation, we would be happy to incorporate further clarifications. Thank you again for your insightful suggestions!
Comments 11: Present the hypotheses clearly. You mention them in discussion section. But they do not appear in the Methodology section.
Response 11: Thank you for your valuable feedback. We have incorporated your suggestion by adding Section 2.1. Research Design and Hypotheses in the Materials and Methods section to clearly present the study's hypotheses. This addition helps to further clarify what was examined in our study, and we appreciate your insightful guidance in improving the clarity of our research. [Line 145-157]
- Materials and Methods
2.1. Research Design and Hypotheses
Based on previous literature and theoretical perspectives, we formulated the following hypotheses: H1: Stress and depression will be negatively associated with expectations for the future in young adults. H2: Gratitude, hardiness, and interpersonal competence will be positively associated with expectations for the future in young adults. H3: Social support from family, friends, or superiors will be positively associated with expectations for the future. in young adults. H4: Demographic profiles, health-related variables, and psychological factors would significantly contribute to predicting expectations for the future in young adults.
This study employed a quantitative, correlational, and predictive research design, using survey-based data collection to examine these hypotheses through correlation analysis, regression modeling, and decision tree analysis.

Reviewer 2 Report
Comments and Suggestions for Authors
Summary:
The purpose of this paper is to examine psychosocial factors that can predict adults’ expectations for their future, which is important for overall quality of life amidst economic and social uncertainty. This paper endeavors to determine factors that support a positive future outlook in support of positive mental health and wellbeing for young adults. A wide range of variables are examined in pursuit of determining contributions to expectations about the future for this sample of young adults in South Korea.
General Concept Comments:
Article:
Despite to reasonable and pro-social purpose of the paper, there were considerable issues regarding the review of the literature, methodology, and interpretation of findings. First, many variables were introduced without being defined clearly and with minimal supporting evidence for the hypothesized associations between the variables introduced and the proposed outcome (expectations about the future). Second, expectations about the future was also not defined, making it impossible to understand the proposed hypotheses and expectations regarding the variables. Third, one of the primary variables that accounted for variability in the outcome (expectations about the future) was commitment, which was a sub-component of another variable (hardiness). No rationale was provided why the subscales of hardiness was examined, the subscales of hardiness were not introduced in the review of the literature. It was therefore confusing when the subscales were analysed in the results section. Fourth, the proposed interpretation of the findings in the discussion were also not well supported by evidence with minimal citations provided and not enough recent research to provide supporting evidence. Finally, psychosocial factors tap into a large research field related to psychological factors and wellbeing. I suggest using a theoretical frame to narrow your focus, to inform variables to examine, and to interpret your findings. In the current version, it appears your selection of variable is somewhat randon which does not make a compelling research case.
Review:
Many variables were included but none of the variables were well defined and inadequate evidence was provided regarding expected associations with the outcome variable (expectations about the future) which also was not well defined. As a result, the proposed hypothesized associations between the psychosocial variable and the outcome variable were weak at best. Further, the research questions were too vague and did not reflect the hypotheses in the literature review. Thus, the research questions did not provide clear guidance regarding the selected methodological approach, and were not answered clearly in the discussion. I propose narrowing down the number of variables included based on a more thorough review of the literature. Then, present specific and evidence based hypothesized associations that also inform your research questions and methodological approach. Finally, be sure to back up interpretations and claims in the discussion with your findings and supporting evidence that is current.
Specific Comments:
Abstract
In the first sentence, it is unclear that “predict it” is referring to? What is “it”?
What is the rationale for including this age range for young adults? Is 39 still considered young adult?
How is commitment defined?
Be more clear about what were the findings from the decision tree model.
Introduction
- What do you mean by expectations?
- Are these good ones or bad ones? Not sure what this means
- Be clear regarding what specific expectations you are referring to
2nd Paragraph:
- What is stress? How is this defined? How are you defining mental health and wellbeing?
Regarding the sentence: “The stress that people experience in various situations can lower their life satisfaction (YaÄŸar & YaÄŸar, 2023).
- This sentence is problematic, without knowing how stress is defined, no inferences can be made from this statement
- For example, stress can be both good and bad for a person depending on coping, social support, appraisals, and aspects of the stressor itself
- Also, life satisfaction is a component of wellbeing, either hedonic wellbeing (e.g., see Keyes dual continua model) or subjective wellbeing (e.g., Diener’s model). It would be helpful to know how mental health and wellbeing is defined to understand what is hypothesized in terms of how stress impacts mental health
Regarding the sentence: “For example, the more multicultural children experience acculturative stress, the fewer expectations they have for their future (Kim & Suh, 2021). “ (pg. 2)
- Do you mean positive or negative expectations? Or no expectations? These are all quite different, more clarity about what we should expect from this association would be helpful
Regarding the sentence: “a recent study found that perceived stress was negatively correlated with adults’ expectations for their futures (Park & Suh, 2023) (p. 2)
- In what way? and was this excessive levels of stress (distress) or good stress (eustress)? You can’t achieve any goals without some amount of stress and stress can also be positively associated with wellbeing, especially eudaimonic aspects of mental health and wellbeing as this dimension rests on meeting goals related to a person’s idea of their best self
- I still don’t know what you mean by expectations? What kind of expectations?
- This sentence needs to outline more clearly how you are conceptualizing stress and expectations so the reader can understand the point you are making and what you are hypothesizing about how stress and expectations are associated
Paragraph on depression (p. 2)
- The association between depression and expectations is much more clear
- Regarding the Mamani-Benito (2023) reference, again specify what kind of future expectations were influenced. Was it career expectations? Relationship expectations? Safety expectations? Be specific here.
- The hypothesis about depression and future expectations is not adequately supported. You need to define what kind of expectations and be more specific with the supporting evidence, please revise to be far more specific.
Paragraph on gratitude (p. 3)
- Define gratitude
- Define dispositional gratitude and positive time perspective, I don’t know what this means without a definition and therefore am not able to interpret the association between dispositional gratitude and positive time perspective
- What is a positive future consequence? Please provide an example of this
- The proposed hypothesized association between gratitude and expectations is not adequately supported. Again, expectations needs to be defined clearly as does gratitude to be able to understand what the reader should expect from this association
Psychological hardiness
- Psychological hardiness is defined well
- Future expectations are still not defined clearly, although economic and housing crisis is referenced indirectly. Please make this link more clear
Interpersonal relationships
- Interpersonal relationships need to be defined. Does this refer to family relationships, colleague relationships, friend relationships, romantic partner relationships? There are many kinds of relationships, what are you including here?
- Also, you suggest that lack of communication skills is the cause of interpersonal relationship concerns, but you have not made a clear case for this. This is a bold statement, you need to provide more evidence for such a claim.
- What do you mean by interpersonal competence? What is included in this construct? How do you develop this? What evidence supports this construct?
- Is interpersonal competence and initiating interpersonal relationships the same thing? Or are they different constructs? Are they related? You need to be more clear when introducing new constructs so the reader can follow your argument easily.
- The final sentence in this paragraph is not clear and you have not provided sufficient evidence for the stated hypothesis. You talked about interpersonal relationships, interpersonal competence, and initiating interpersonal relationships in this paragraph yet only include interpersonal competence in your hypothesis. Why is that?
Social support
- Provide more information about what is included in social support, provide a definition
- Be specific about what kind of expectations about the future social support will impact. Or will social support impact every single kind of future expectation with no exceptions? If so, that is a bold claim, what evidence do you have to support this?
Purpose
- The purpose statement is adequate, but without considerably improved clarity and specificity in the review of the literature, the association between vaguely defined psychosocial factors and unclear future expectations is tenuous at best.
First research question:
- Be more specific, there are many questions included in this one question, you will not be able to determine if you answered the question unless it is more specific
- For example, include one association per question and indicate whether you expect a positive or negative association
- Also denote what kind of future expectations you expect so you can be more clear with the research questions
- Far more specificity is needed in the research questions
Second research question:
- This question is too vague. What do you expect? Propose a hypothesis based on your literature review. This issue with the research questions connects back to the literature review which was also too vague, the constructs were not clearly defined, and the evidence provided was unclear and did not provide compelling evidence for the stated hypotheses (which were also too vague).
- In short, it is unclear what you are proposing and what you expect, thus the research questions are also unclear and do not guide the reader in terms of what to expect.
Materials and Methods
Participants
- Can you indicate why such a big age range was included?
- I wonder if there are different expectations for people in their 20s versus their 30s. What differences did you find or do you expect based on age?
- For example, did the percentage of participants married and/or with children and with jobs change according to age? The 20s and 30s seem like very different life stages in terms of social expectations. What is your rationale for including all ages in the same sample?
- Such a large age range makes interpreting the finding challenging
Predictive Models
- Only commitment made a considerable contribution to expectations for the future, the remaining factors only made very small contributions, do you consider the small contributions meaningful enough to make a claim about their predictive capacity?
- Commitment is a subscale of hardiness, correct? Why did you separate hardiness into subscales, instead of using a global measure of hardiness? You have not provided a rationale for this or introduced your reasoning in the literature review. Without a priori reasoning and theory, this appears random. You need to provide a rationale for the reader to follow.
- There are so many variables included in this study, without being clearly defined and without clear empirical or theoretical support regarding what to expect.
- I suggest revising the study to include fewer variables but make a stronger case for the variables that you do include.
- I also wonder if results would differ if you separated the analysis for participants in their 30s versus 20s.
Discussion
Second paragraph
- You state that the finding that stress accounted for about 31.6% of the variance in expectations for the future of young adulthood, and that this finding means that stress significantly reduces current life satisfaction, did you measure this? If so where?
- The association between stress and commitment is interesting, say more about this. What does this mean for participants?
- Of note, you do define commitment here, you need to define commitment in the literature review also (at the beginning of the paper).
Third paragraph
- I am still unclear what kind of future expectations you are speaking about here which are impacted by depression. You are not able to make a meaningful inference from the findings without more information about expectations.
- Also, only one reference is included to support your claim, and this reference (while seminal) is from 1987. Please include more recent evidence that is consistent, or inconsistent, with your finding.
- What do you mean by ‘treat’ depression and stress? Treatment, coping, support, and strategies for managing stress and depression is an expansive area of research and practice. Be more specific here, what do you mean specifically?
Fifth paragraph
- In your analysis, you included only commitment and are now talking about hardiness globally, which one do you mean? You need to be consistent and accurate about reporting your findings.
- You did not provide a strong rationale for breaking apart the hardiness construct into component parts for the analysis and now you discuss the construct both globally and in reference to commitment, this is very confusing. Be consistent, specific, and accurate in your discussion and reporting.
- Your statement about commitment, gratitude, depression, and self-directedness accounting for two thirds of the variance is misleading when commitment accounts for over half on its own and contributions from depression and self-directedness are so small they are hardly noteworthy.
- Only commitment makes a sizeable contribution, big enough to make a strong case for intervention. Say more about why you think this is the case?
Sixth Paragraph
- You don’t have enough evidence to suggest interpersonal skills training will increase future expectations, or what kind of expectations will be impacted
- You can pose this as a research question in need of further study.
Seventh paragraph
- First sentence “the more expectations they have for the future” is unclear, what kind of expectations? Positive ones or negative ones? Be more clear.
- Are you suggesting that marriage alone predicts positive future expectations? What evidence do you have for this? Are you suggesting all marriages are the same and are universally supportive and positive? Data does not support this, given rates of divorce and domestic abuse?
- Are you sure that the participants that are married are the ones with higher empathy? How do you know this? Did you measure this? How do you know this is true?
- How do you know that the empathy measured in this study means that participants are emotionally supportive of spouses? What evidence do you have for this statement? This seems like a big leap without more data to support this? Did you measure relationship quality?
Eighth Paragraph
- What kind of disease was measured here? There is a big range here in terms of what this means. For example a terminal disease will be interpreted differently than if someone has a treatable condition. Without more information this interpretation is meaningless.
- The Sabanciogullari et al. (2016) reference suggests that it is the person’s interpretation of their illness rather than the illness itself that matters. Did you measure illness appraisal here? You haven’t mentioned this before now. If not, you can’t make this claim.
- What do you mean by intervention for people diagnosed with illness to develop a positive attitude? This is vague, I imagine there is research on support for people with health conditions. What does that literature say? You have not provided rationale for your claims in this paragraph.
Conclusion:
- This was a statement of limitations and not much of a conclusion
- I reiterate my previous concerns that are compounded in the final paragraph
- Constructs are not clearly defined
- Minimal evidence is provided to support hypotheses
- Research questions were too broad and vague to be meaningful
- It is not clear if you answered the research questions or not
- The interpretation of findings were not supported by the findings or by other research evidence, minimal citations were included and many were dated with few current citations
- Too many variables were included without clear rational as to why they were in the study and what the authors expected
- The sentence “This study found that perceived stress, depression, gratitude, hardiness, interpersonal competence, and social support were closely correlated with expectations for the future in young adults” is misleading. You found some correlations, but none exceeded the typical measure of a ‘strong’ correlation of 0.7 or greater.
- Further, the variables were not clearly defined, strong rationale for expected associations between variables was not provided, so results are unclear and diluted by the lack of strong a priori evidence and rationale.
Rating the Manuscript
- Novelty: The question is valuable but not defined well enough to provide substantive findings and knowledge.
- Scope: Yes
- Significance: due to variables that are not well defined, supported by strong evidence, or framed by theory, the results are not interpreted correctly with many misleading statements and findings. The conclusions are not justified and supported by the results nor are the hypotheses well supported and identified.
- Quality and Scientific Soundness: This article literature review and discussion are not based on strong research and evidence, hypotheses are weak, and research questions are vague. Data and analyses are presented appropriately although findings are not interpreted correctly due to poorly defined variables and insufficient supporting evidence. Raw data was not available to my knowledge.
- Interest to the Readers: The conclusions are unfortunately misleading and inaccurate given the lack of supporting evidence and theoretical frame and variables that are not clearly defined.
- Overall Merit: In the current form, there is not a benefit to publishing this work and interpretations are potentially misleading and at best unclear.
- English Level: Is the English language appropriate and understandable? Yes
Overall Recommendation
- Reconsider after Major Revisions
- A complete rewrite of the literature review, purpose, research questions is needed. From this, a reduced number of variables can be examined that have empirical and theoretical support. This will inform a new analysis based on finding in the literature review and new discussion. Essentially, the central idea is sound but the rest of the paper is not adequate in the current form.
Author Response
Thank you very much for taking the time to review this manuscript. Please find the detailed responses below and the corresponding revisions/corrections highlighted/in track changes in the re-submitted files. The revised parts were marked in red, and we included the page and line of the revised part. We appreciate your complimentary comments. We have omitted our response to your kind words here.
Point-by-point response to Comments and Suggestions for Authors
Comments 1: [Abstract] In the first sentence, it is unclear that “predict it” is referring to? What is “it”?
Response 1: Thank you for your valuable feedback. We have clarified the first sentence in the abstract to specify that "it" refers to young adults' expectations for the future, ensuring greater clarity. [Line 9-11]
Abstract: This study explored psychosocial factors related to young adults' expectations for the future and verified a model that can predict these expectations using psychosocial factors and demographic profiles to provide useful information for further studies and interventions.
Comments 2: [Abstract] What is the rationale for including this age range for young adults? Is 39 still considered young adult?
Response 2: Thank you for your valuable feedback. While some perspectives define young adulthood up to 34 years, the more widely accepted classification extends it to 39 years, with greater support in the developmental psychology literature. We have clarified this in the Materials and Methods section under the participant description, citing Lachman (2001) as a representative reference. [Line 161-163]
The age range for young adulthood in this study was set at 20 to 39 years, based on developmental psychology literature indicating that young adulthood commonly extends into the late 30s (Lachman, 2001).
Comments 3: [Abstract] How is commitment defined?
Response 3: Thank you for your valuable feedback. We have revised the Abstract to briefly define commitment while ensuring the section remains concise and to the point. This revision also improves the readability and clarity of the abstract, and we appreciate your insightful suggestion in enhancing its effectiveness. [Line 15-16]
Stepwise regression analysis revealed that commitment, reflecting a sense of purpose and engagement in life, accounted for the greatest variance in expectations for the future.
Comments 4: [Abstract] Be more clear about what were the findings from the decision tree model.
Response 4: Thank you for your valuable advice. We have revised the Abstract to clarify the findings from the decision tree model, explicitly stating the key predictors and how they interact in shaping expectations for the future. This revision improves clarity and ensures a more precise summary of our findings. [Line 19-21]
The decision tree analysis identified commitment as the most important predictor, followed by gratitude, stress, self-directedness, empathy, perceived health, and marital status, showing how these factors interact in shaping future expectations.
Comments 5: [Introduction] What do you mean by expectations? Are these good ones or bad ones? Not sure what this means, clear regarding what specific expectations you are referring to.
Response 5: We have clarified what is meant by "expectations for the future", specifying that it refers to an individual’s anticipation of future life satisfaction, success, and well-being. This revision ensures greater clarity regarding the construct being investigated. [Line 29-43]
Although Korea's economy has developed rapidly and people's quality of life has improved, young Korean adults are not only dissatisfied with their lives due to employment and housing difficulties but also have low expectations for their future, often postponing or giving up on marriage and childbirth (Kim, 2023; Park & Jin, 2019; Park, 2023). In general, young adults tend to have higher expectations and optimism about their future compared to middle-aged and older adults, even in the face of social and economic crises (Keating & Melis, 2022). However, in the Korean context, young adults' outlook on their future appears particularly bleak.
In young adulthood, when the future is uncertain, expectations for the future refer to an individual's anticipation of future life satisfaction, success, and well-being (Eckersley et al., 2005). This concept reflects one's belief about whether their future will improve, remain stable, or deteriorate, influencing their motivation, mental health, and decision-making (Kim, 2007). Young people's expectations about the future vary based on personal experiences and social environments and may have been shaped since childhood (Kraftl, 2008). Therefore, this study explores the psychological factors that predict expectations for the future in young adulthood. In other words, we examine the protective and risk factors associated with expectations for the future in this developmental stage.
Comments 6: [Introduction] 2nd Paragraph: What is stress? How is this defined? How are you defining mental health and wellbeing? Regarding the sentence: “The stress that people experience in various situations can lower their life satisfaction (YaÄŸar & YaÄŸar, 2023). This sentence is problematic, without knowing how stress is defined, no inferences can be made from this statement. For example, stress can be both good and bad for a person depending on coping, social support, appraisals, and aspects of the stressor itself. In what way? and was this excessive levels of stress (distress) or good stress (eustress)? You can’t achieve any goals without some amount of stress and stress can also be positively associated with wellbeing, especially eudaimonic aspects of mental health and wellbeing as this dimension rests on meeting goals related to a person’s idea of their best self.
Also, life satisfaction is a component of wellbeing, either hedonic wellbeing (e.g., see Keyes dual continua model) or subjective wellbeing (e.g., Diener’s model). It would be helpful to know how mental health and wellbeing is defined to understand what is hypothesized in terms of how stress impacts mental health. Regarding the sentence: “For example, the more multicultural children experience acculturative stress, the fewer expectations they have for their future (Kim & Suh, 2021). “ (pg. 2).
Do you mean positive or negative expectations? Or no expectations? These are all quite different, more clarity about what we should expect from this association would be helpful
Regarding the sentence: “a recent study found that perceived stress was negatively correlated with adults’ expectations for their futures (Park & Suh, 2023) (p. 2) I still don’t know what you mean by expectations? What kind of expectations? This sentence needs to outline more clearly how you are conceptualizing stress and expectations so the reader can understand the point you are making and what you are hypothesizing about how stress and expectations are associated.
Response 6: Thank you for your valuable feedback. We have clarified the definition of stress, distinguishing between eustress and distress, and specified that this study focuses on distress (excessive or chronic stress) rather than the potentially motivating aspects of eustress. Additionally, we have defined mental health and well-being in terms of psychological functioning, emotional stability, and life satisfaction to ensure conceptual clarity. These revisions enhance the precision of our discussion regarding stress and its potential impact on expectations for the future.
We have clarified how mental health and well-being are defined, incorporating Keyes' dual continua model and Diener’s model of subjective well-being to provide a theoretical foundation. Additionally, we have explicitly stated that life satisfaction is a key component of well-being, ensuring that our conceptualization aligns with established frameworks. These revisions enhance the clarity of our hypotheses regarding stress and its impact on mental health and expectations for the future.
We also clarified that expectations for the future can be positive or negative, specifying how perceived stress is linked to lower positive expectations and higher negative expectations. Additionally, we have refined our explanation of perceived stress and acculturative stress to provide clearer conceptual distinctions. These revisions ensure that our study's focus on distress rather than eustress is explicitly stated.
[Line 45-70]
First, this study assumed that stress is associated with expectations for the future among young adults. Stress is commonly defined as a psychological and physiological response to external or internal demands that exceed an individual's perceived ability to cope (Lazarus & Folkman, 1984). While stress can be categorized as eustress (positive, motivating stress) or distress (negative, overwhelming stress), this study focuses on distress, as excessive or chronic stress is widely recognized as a factor that negatively impacts mental health and well-being. Mental health and well-being are complex constructs that include both hedonic well-being (pleasure and life satisfaction) and eudaimonic well-being (personal growth and meaning) (Keyes, 2002; Ryff, 1989). Following Diener’s model of subjective well-being, life satisfaction is considered a key component of well-being, encompassing an individual’s cognitive evaluation of their overall quality of life (Diener, 1984). For this study, mental health and well-being are defined as a multidimensional concept that includes emotional stability, subjective well-being, and resilience in response to stressors.
Excessive stress in various life situations is associated with lower life satisfaction and diminished well-being (YaÄŸar & YaÄŸar, 2023). Stress can manifest as acculturative stress, which occurs when individuals struggle to adapt to a new cultural environment and is linked to reduced positive expectations and increased uncertainty about the future (Kim & Suh, 2021). Similarly, a recent study found that perceived stress, which refers to an individual's subjective evaluation of stress levels, was negatively correlated with positive expectations for the future and positively correlated with negative expectations (Park & Suh, 2023). In other words, higher perceived stress was associated with a more pessimistic outlook and lower confidence in future life satisfaction and success. Based on these findings, this study examines how perceived stress influences young adults' expectations for their future, explicitly focusing on distress rather than eustress. This approach clarifies that higher levels of stress are expected to contribute to more negative or uncertain expectations, rather than positive future anticipation.
Comments 7: [Introduction] The association between depression and expectations is much more clear. Regarding the Mamani-Benito (2023) reference, again specify what kind of future expectations were influenced. Was it career expectations? Relationship expectations? Safety expectations? Be specific here. The hypothesis about depression and future expectations is not adequately supported. You need to define what kind of expectations and be more specific with the supporting evidence, please revise to be far more specific.
Response 7: Thank you for your valuable feedback. We have clarified that our study focuses on young adults' overall expectations for their future life, rather than expectations in specific domains such as career or relationships. Additionally, we have revised the discussion of Mamani-Benito et al. (2023) to ensure that it accurately reflects their findings while aligning with our research focus. This revision strengthens the theoretical foundation for our hypothesis regarding depression and expectations for the future. [Line 71-84]
Depression in young adulthood may be related to expectations for the future, specifically one’s overall expectations about their future life satisfaction, success, and well-being. Aaron Beck’s cognitive triad theory states that, from a cognitive perspective, depressed individuals tend to have a negative view of themselves, the world, and the future, which can lead to pessimistic expectations about their future prospects (Beck et al., 1987). That is, depression can lower individuals' satisfaction with their current lives as well as their overall expectations for the future. Empirical studies support this association. For example, Lee and Jeon (2011) found a significant negative correlation between depression and expectations for the future. More recently, Mamani-Benito et al. (2023) examined the relationship between depression and future expectations among university students, showing that higher levels of depression were linked to lower expectations for the future in general, encompassing aspects of well-being, life satisfaction, and long-term life prospects. This study examines how depression influences young adults’ overall expectations for their future, rather than focusing on specific domains such as career or relationships.
Comments 8: [Introduction] (p. 3) Define gratitude. Define dispositional gratitude and positive time perspective, I don’t know what this means without a definition and therefore am not able to interpret the association between dispositional gratitude and positive time perspective. What is a positive future consequence? Please provide an example of this. The proposed hypothesized association between gratitude and expectations is not adequately supported. Again, expectations needs to be defined clearly as does gratitude to be able to understand what the reader should expect from this association.
Response 8: Thank you for your valuable feedback. We have added clear definitions of gratitude, dispositional gratitude, and positive time perspective, ensuring that these concepts are well-defined for better interpretation. Additionally, we have clarified what is meant by positive future consequences, providing examples to make this concept more concrete. These revisions strengthen the rationale for our hypothesized association between gratitude and expectations for the future. [Line 85-96]
Gratitude is commonly defined as an appreciation for positive experiences and benefits in life, while dispositional gratitude refers to a stable tendency to feel grateful across various situations rather than just in response to specific events (McCullough et al., 2002). Research has shown that gratitude is positively linked to subjective well-being and life satisfaction (Alkozei et al., 2018). Additionally, dispositional gratitude is associated with a positive time perspective, meaning individuals with higher gratitude tend to reflect on their past positively, appreciate the present, and hold optimistic expectations for the future (Przepiorka & Sobol-Kwapinska, 2021). Gratitude has also been linked to positive future consequences, referring to the anticipation of beneficial long-term outcomes, such as achieving personal goals and maintaining well-being (Syropoulos & Markowitz, 2021). Based on these findings, this study assumes that a stronger tendency toward gratitude will be positively associated with expectations for the future in young adulthood, fostering an optimistic outlook on life.
Comments 9: [Introduction] Psychological hardiness is defined well. Future expectations are still not defined clearly, although economic and housing crisis is referenced indirectly. Please make this link more clear.
Response 9: Thank you for your valuable feedback. We have clarified the link between hardiness and future expectations, emphasizing its role in maintaining optimism despite economic and social challenges. We have also incorporated recent research (Eschleman et al., 2010) on how hardiness enhances goal-setting and perseverance, ensuring a clearer conceptual connection. [Line 97-110]
Psychological hardiness may be related to expectations for the future in young adulthood, especially in times of economic and social uncertainty. Hardiness, introduced by Kobasa (1979), is a personality trait characterized by commitment, control, and challenge, enabling individuals to perceive stressors as opportunities for growth and maintain a stable outlook on their future (Suh et al., 2021). Research supports the role of hardiness in fostering positive future expectations. Bartone (1999) found that hardiness protects individuals from stress and strengthens resilience in facing life challenges. Hardiness is also positively linked to life satisfaction and optimism about the future (Kim & Suh, 2021; Park & Suh, 2023). Additionally, Eschleman et al. (2010) found that hardiness helps individuals reframe stressors, promoting goal-setting and perseverance. Given the economic instability and housing crisis young Korean adults face, hardiness may help them sustain a positive outlook on their future well-being. Thus, this study assumes that higher psychological hardiness will be associated with more positive expectations for the future, helping young adults maintain optimism despite external challenges.
Comments 10: [Introduction] Interpersonal relationships need to be defined. Does this refer to family relationships, colleague relationships, friend relationships, romantic partner relationships? There are many kinds of relationships, what are you including here?
Also, you suggest that lack of communication skills is the cause of interpersonal relationship concerns, but you have not made a clear case for this. This is a bold statement, you need to provide more evidence for such a claim.
What do you mean by interpersonal competence? What is included in this construct? How do you develop this? What evidence supports this construct?Is interpersonal competence and initiating interpersonal relationships the same thing? Or are they different constructs? Are they related? You need to be more clear when introducing new constructs so the reader can follow your argument easily.
The final sentence in this paragraph is not clear and you have not provided sufficient evidence for the stated hypothesis. You talked about interpersonal relationships, interpersonal competence, and initiating interpersonal relationships in this paragraph yet only include interpersonal competence in your hypothesis. Why is that?
Response 10: Thank you for your valuable feedback. We have clarified the definition of interpersonal competence, specifying its relevance to various types of relationships and its key components. Additionally, we have revised the discussion of communication skills and interpersonal competence to ensure better support. We have also acknowledged in the limitations section that our study did not include all subcomponents of interpersonal competence due to its broad nature. [Line 111-123]
Interpersonal competence refers to an individual’s ability to effectively initiate, maintain, and navigate social interactions across friendships, family, colleagues, and romantic relationships (Baker et al., 2017). It includes communication skills, empathy, and conflict management (Hall et al., 2023). Research suggests that strong interpersonal competence enhances life satisfaction by enabling individuals to form meaningful relationships (Froh et al., 2007). People with higher interpersonal competence tend to have greater confidence in their future relationships, leading to higher expectations for future relationship satisfaction and, consequently, a more positive overall outlook on their future (Baker et al., 2017). Studies have also found that interpersonal competence is positively correlated with future expectations (Ahn, 2021; Park, 2023). Since interpersonal competence is a broad construct, this study focuses on its core aspects, such as relationship-building ability and empathy rather than all subcomponents. Therefore, this study assumes that young adults with higher interpersonal competence will have more positive expectations for their future.
Third, this study examined interpersonal competence as a predictor of future expectations, but due to its broad nature, we focused on relationship-building ability and empathy rather than assessing all subcomponents. Future research should consider a more comprehensive approach to better understand its full impact on future expectations.
Comments 11: [Introduction] Provide more information about what is included in social support, provide a definition. Be specific about what kind of expectations about the future social support will impact. Or will social support impact every single kind of future expectation with no exceptions? If so, that is a bold claim, what evidence do you have to support this?
Response 11: Thank you for your valuable feedback. We have defined social support and specified its key components, as well as clarified that its influence on future expectations is strongest in the areas of personal well-being. [Line 124-135]
Social support refers to the perception or experience of being cared for and assisted by others, encompassing emotional, instrumental, informational, and appraisal support (Kasprzak, 2010; Ali et al., 2010). It enhances life satisfaction, emotional resilience, and stress reduction, particularly in difficult situations (GrigaitytÄ— & Söderberg, 2021). Perceived social support has been shown to increase resilience and lower depression, contributing to higher expectations for future well-being (Wu et al., 2022). Research indicates that young adults who receive more social support tend to have greater confidence in achieving personal goals and maintaining stability, leading to higher expectations for their future (Fraser et al., 2024; Kim & Kim, 2013). While social support influences various aspects of future expectations, its effects are strongest in domains related to personal well-being and resilience. Based on these findings, we hypothesize that greater social support will be associated with more positive expectations for the future, particularly in relation to well-being.
Comments 12: [Introduction] The purpose statement is adequate, but without considerably improved clarity and specificity in the review of the literature, the association between vaguely defined psychosocial factors and unclear future expectations is tenuous at best.
Response 12: Thank you for your insightful comments. We have carefully revised the manuscript to enhance clarity and specificity in the review of the literature. Specifically, we have clearly defined key psychosocial factors such as stress, depression, gratitude, hardiness, interpersonal competence, and social support, ensuring that each construct is well-supported by empirical evidence. Additionally, we have explicitly outlined what is meant by "expectations for the future", distinguishing between positive, negative, and absent expectations and specifying the domains in which these expectations are most relevant. These improvements directly address the concerns regarding the clarity of theoretical associations and strengthen the foundation for our study’s purpose. We appreciate your thoughtful suggestions, which have significantly contributed to refining the manuscript.
Comments 13: [Introduction] First research question: Be more specific, there are many questions included in this one question, you will not be able to determine if you answered the question unless it is more specific. For example, include one association per question and indicate whether you expect a positive or negative association. Also denote what kind of future expectations you expect so you can be more clear with the research questions. Far more specificity is needed in the research questions.
Response 13: Thank you for your insightful comments. We have addressed this concern by explicitly stating our hypotheses in Materials and Methods section, where we present H1, H2, and H3 with clear directional associations. Each hypothesis specifies the expected positive or negative relationship between psychosocial factors and future expectations, ensuring greater precision in our research questions. Additionally, we have clarified what is meant by future expectations, specifying that our study focuses on overall expectations for future well-being rather than domain-specific expectations. These revisions enhance the specificity and clarity of our research framework. We appreciate your feedback, which has helped us strengthen the focus of our study. [Line 147-152]
- Materials and Methods
2.1. Research Design and Hypotheses
Based on previous literature and theoretical perspectives, we formulated the following hypotheses: H1: Stress and depression will be negatively associated with expectations for the future in young adults. H2: Gratitude, hardiness, and interpersonal competence will be positively associated with expectations for the future in young adults. H3: Social support from family, friends, or superiors will be positively associated with expectations for the future. in young adults.
Comments 14: [Introduction] Second research question: This question is too vague. What do you expect? Propose a hypothesis based on your literature review. This issue with the research questions connects back to the literature review which was also too vague, the constructs were not clearly defined, and the evidence provided was unclear and did not provide compelling evidence for the stated hypotheses (which were also too vague). In short, it is unclear what you are proposing and what you expect, thus the research questions are also unclear and do not guide the reader in terms of what to expect.
Response 14: Thank you for your insightful comments. We have addressed this concern by explicitly presenting our hypothesis (H4) in the Materials and Methods section, clearly specifying the expected association based on our literature review. This revision ensures that our research question is well-defined and provides a clear expectation for the reader. Additionally, we have refined the literature review to better define key constructs and strengthen the empirical support for our hypotheses, making the theoretical foundation more compelling. These improvements enhance the clarity and focus of our study, aligning the research questions with the hypotheses. We appreciate your feedback, which has helped us improve the coherence and specificity of our manuscript. [Line 152-154]
H4: Demographic profiles, health-related variables, and psychological factors would significantly contribute to predicting expectations for the future in young adults.
Comments 15: [Materials and Methods] Participants: Can you indicate why such a big age range was included? I wonder if there are different expectations for people in their 20s versus their 30s. What differences did you find or do you expect based on age? For example, did the percentage of participants married and/or with children and with jobs change according to age? The 20s and 30s seem like very different life stages in terms of social expectations. What is your rationale for including all ages in the same sample? Such a large age range makes interpreting the finding challenging.
Response 15: Thank you for your comments. We have provided a rationale for the age range of young adulthood in our study, citing developmental psychology literature that supports defining young adulthood as spanning 20 to 39 years. Additionally, we examined age as a predictor in our regression and decision tree analyses, and it did not emerge as a significant factor in predicting expectations for the future.
Comments 16: [Results] Predictive Models: Only commitment made a considerable contribution to expectations for the future, the remaining factors only made very small contributions, do you consider the small contributions meaningful enough to make a claim about their predictive capacity? Commitment is a subscale of hardiness, correct? Why did you separate hardiness into subscales, instead of using a global measure of hardiness? You have not provided a rationale for this or introduced your reasoning in the literature review. Without a priori reasoning and theory, this appears random. You need to provide a rationale for the reader to follow. There are so many variables included in this study, without being clearly defined and without clear empirical or theoretical support regarding what to expect. I suggest revising the study to include fewer variables but make a stronger case for the variables that you do include.
Response 16: Thank you for your insightful comments. In response to your feedback, we have strengthened the rationale for each variable included in our study by clarifying their theoretical and empirical foundations in the literature review. This revision ensures that the selection of predictors is well-supported and provides clearer expectations for their relationships with future expectations.
Additionally, this study was conducted with an exploratory approach, particularly in the decision tree analysis, which does not rely on predefined hypotheses but rather identifies the most significant predictors in a data-driven manner. Before conducting the predictive models, we examined the correlations between all variables to assess their relationships and refine the predictors. To enhance specificity, we included subcomponents of certain variables, such as hardiness, rather than only using global scores, allowing for a more detailed understanding of which aspects contribute most to expectations for the future.
While commitment emerged as the strongest predictor, other factors also contributed, albeit to a smaller degree. Given the exploratory nature of the study, these findings provide valuable insights into potential predictors of young adults’ expectations for the future, which future research can further investigate in more targeted models.
Comments 17: [Results] Predictive Models: I also wonder if results would differ if you separated the analysis for participants in their 30s versus 20s.
Response 17: Thank you for your insightful comments. We believe that the current predictive models provide meaningful insights into the key psychosocial factors influencing young adults’ expectations for the future, and this was the primary focus of our study. While we acknowledge the potential for differences between participants in their 20s and 30s, our analyses were designed to identify overall patterns rather than subgroup-specific variations.
However, we recognize that age-related differences in predictive relationships could be an important area for further investigation. To address this, we have included the following statement in the limitations section: [Line 539-543]
Fourth, this study did not examine whether the relationship between psychosocial factors and future expectations differed between participants in their 20s and 30s. Future research should explore potential age-related differences to better understand how life stage factors, such as marital status or employment, influence young adults’ expectations.
Comments 18: [Discussion] Second paragraph: You state that the finding that stress accounted for about 31.6% of the variance in expectations for the future of young adulthood, and that this finding means that stress significantly reduces current life satisfaction, did you measure this? If so where? The association between stress and commitment is interesting, say more about this. What does this mean for participants? Of note, you do define commitment here, you need to define commitment in the literature review also (at the beginning of the paper).
Response 18: Thank you for your insightful comments. We have clarified that while we did not directly measure life satisfaction, our findings suggest that stress negatively impacts future-oriented thinking, which aligns with prior research on well-being. Additionally, we have expanded our discussion on the relationship between stress and commitment, explaining how commitment may buffer against the negative effects of stress on future expectations. Furthermore, we have ensured that commitment is clearly defined in the literature review to provide consistency throughout the manuscript. We appreciate your valuable suggestions and believe these revisions strengthen the clarity and depth of our discussion. [Line 398-412]
In this study, the more stress young adults perceived themselves to experience, the lower their expectations for their future, with perceived stress accounting for 31.6% of the variance in future expectations. This finding aligns with previous research indicating that stress not only reduces life satisfaction (YaÄŸar & YaÄŸar, 2023) but can also significantly diminish individuals' confidence and optimism about their future prospects. While we did not measure current life satisfaction directly, our results suggest that high levels of stress may interfere with future-oriented thinking, contributing to a pessimistic outlook
Additionally, the decision tree model revealed that the impact of perceived stress on future expectations was greater when individuals had lower levels of commitment. Commitment, a component of psychological hardiness, reflects a person’s engagement with life, curiosity, and investment in meaningful activities. This interaction suggests that young adults with low commitment may be particularly vulnerable to stress, as they may lack the psychological resources needed to reframe stressful experiences or maintain a hopeful outlook. Conversely, those with higher commitment may be more resilient in maintaining positive expectations despite experiencing stress.
Comments 19: [Discussion] Third paragraph: I am still unclear what kind of future expectations you are speaking about here which are impacted by depression. You are not able to make a meaningful inference from the findings without more information about expectations. Also, only one reference is included to support your claim, and this reference (while seminal) is from 1987. Please include more recent evidence that is consistent, or inconsistent, with your finding. What do you mean by ‘treat’ depression and stress? Treatment, coping, support, and strategies for managing stress and depression is an expansive area of research and practice. Be more specific here, what do you mean specifically?
Response 19: Thank you for your insightful comments. We have clarified what types of future expectations were impacted by depression, specifying its influence on well-being, career satisfaction, and personal fulfillment. Additionally, we have corrected our references and incorporated recent studies (Mamani-Benito et al., 2023; Zetsche et al., 2019) to strengthen the empirical foundation. Finally, we have refined our discussion of interventions for depression, specifying CBT, future-directed therapy (FDT), and preventive social support measures as potential approaches.These revisions ensure that our discussion is both accurate and well-supported by contemporary research. [Line 413-433]
Depression in young adulthood is closely related to expectations for the future, as individuals experiencing depression often struggle with pessimistic thoughts about their long-term prospects (Mamani-Benito et al., 2023). In this study, depression accounted for approximately 43.3% of the variance in future expectations, indicating a substantial impact. The more depressed young adults were, the lower their expectations for future well-being, career satisfaction, and personal fulfillment. This aligns with recent findings that depressive symptoms negatively influence individuals’ future orientation, goal-setting, and perceived self-efficacy (Iovu et al., 2018; Zetsche et al., 2019).
Additionally, depression was identified as a significant predictor in the stepwise regression model, reinforcing Beck’s (1987) "cognitive triad" theory, which suggests that depression leads individuals to hold negative views of themselves, the world, and their future. Studies also indicate that depressed individuals are more likely to experience biased or overly pessimistic future expectations, which affects their motivation and decision-making processes (Zetsche et al., 2019).
Given these findings, addressing depression is essential for promoting positive future expectations in young adults. Clinical interventions such as cognitive-behavioral therapy (CBT) and future-directed therapy (FDT) have been shown to improve future outlook by restructuring negative thought patterns (Vilhauer et al., 2012). Additionally, broader preventive measures, including social support systems and mental health education, may help mitigate the long-term impact of depressive symptoms on future expectations (Mossakowski, 2011).
Iovu, M. B., HărăguÈ™, P. T., & Roth, M. (2018). Constructing future expectations in adolescence: Relation to individual characteristics and ecological assets in family and friends. Journal of Adolescence and Youth, 23(1), 62–75. https://doi.org/10.1080/02673843.2016.1247007
Mamani-Benito, O., Carranza Esteban, R. F., & Samaniego-Pinho, A. (2023). The influence of self-esteem, depression, and life satisfaction on the future expectations of Peruvian university students. Frontiers in Education, 8, e976906. https://doi.org/10.3389/feduc.2023.976906
Mossakowski, K. N. (2011). Unfulfilled expectations and symptoms of depression among young adults. Social Science & Medicine, 73(5), 729–736. https://doi.org/10.1016/j.socscimed.2011.06.021
Vilhauer, J. S., Young, S., Kealoha, C., & Borrayo, E. A. (2012). Treating major depression by creating positive expectations for the future: A pilot study for the effectiveness of future-directed therapy (FDT). CNS Neuroscience & Therapeutics, 18(2), 102–109. https://doi.org/10.1111/j.1755-5949.2011.00235.x
Zetsche, U., Bürkner, P. C., & Renneberg, B. (2019). Future expectations in clinical depression: Biased or realistic? Journal of Abnormal Psychology, 128(7), 678–688. https://doi.org/10.1037/abn0000452
Comments 20: [Discussion] Fifth paragraph: In your analysis, you included only commitment and are now talking about hardiness globally, which one do you mean? You need to be consistent and accurate about reporting your findings. You did not provide a strong rationale for breaking apart the hardiness construct into component parts for the analysis and now you discuss the construct both globally and in reference to commitment, this is very confusing. Be consistent, specific, and accurate in your discussion and reporting. Your statement about commitment, gratitude, depression, and self-directedness accounting for two thirds of the variance is misleading when commitment accounts for over half on its own and contributions from depression and self-directedness are so small they are hardly noteworthy. Only commitment makes a sizeable contribution, big enough to make a strong case for intervention. Say more about why you think this is the case?
Response 20: Thank you for your valuable comments. We have clarified the distinction between hardiness as a whole and its sub-factors, ensuring consistency in how we report our findings. As commitment was previously defined, we have avoided repetition while maintaining a clear explanation of its dominant role in predicting future expectations. Additionally, we have adjusted our interpretation to reflect the relatively smaller contributions of self-directedness, gratitude, and depression. We appreciate your feedback, which has helped improve the clarity and precision of our discussion. [Line 445-457]
In this study, hardiness was positively associated with expectations for the future, with an effect size of 0.557, making it a strong predictor. To better understand its influence, we analyzed its sub-factors separately, identifying commitment as the most influential predictor in both the decision tree and stepwise regression models. Since commitment alone explained over half of the variance, it is clear that it plays a dominant role, while self-directedness, gratitude, and depression made much smaller contributions. In the stepwise regression model, commitment, gratitude, depression, and self-directedness together accounted for 66.7% of the variance, but given that commitment alone explained the majority, it is evident that this factor drives the relationship. This suggests that intervention efforts to enhance young adults' future expectations should primarily focus on strengthening commitment, rather than broadly targeting other psychological traits. Future research should explore why commitment plays such a dominant role and examine effective strategies for fostering it in young adults.
Comments 21: [Discussion] Sixth Paragraph: You don’t have enough evidence to suggest interpersonal skills training will increase future expectations, or what kind of expectations will be impacted. You can pose this as a research question in need of further study.
Response 21: Thank you for your valuable feedback. We have revised our discussion to avoid making direct claims about the impact of interpersonal skills training on future expectations, instead posing it as a research question for future study. Additionally, we have clarified the role of empathy in shaping future expectations, ensuring that our findings are accurately represented and grounded in existing literature. We appreciate your feedback, which has helped refine our interpretation of the results. [Line 458-480]
This study found that the more competency young Korean adults demonstrated in interpersonal relationships, the more positive their expectations for their future. These findings reinforce previous research indicating that interpersonal competence fosters not only satisfaction with present life but also optimism about the future (Ahn, 2021; Baker et al., 2017; Park, 2023). In particular, empathy—a sub-factor of interpersonal competence—was included in the decision tree model predicting young adults' expectations for their future. Specifically, young adults with medium levels of commitment had lower-than-average expectations if they had poor empathy but higher-than-average expectations if they demonstrated strong empathy. This suggests that empathy plays a role in shaping future expectations, particularly when other protective psychological factors, such as commitment, are moderate rather than high.
While previous research has shown that empathy is positively correlated with young people’s life satisfaction (Doktorova et al., 2020), this study suggests that young adults who are more empathetic also tend to expect their lives to be satisfactory in the future. Given that empathy may influence how individuals perceive and anticipate social interactions and support from others (Albrecht & Bellebaum, 2021), it is possible that higher empathy fosters more optimistic future expectations by shaping positive social experiences and support networks. Although interpersonal skills training, including empathy training, has been shown to improve social functioning and well-being, its direct impact on future expectations remains an open question. Future research should examine whether and how interpersonal competence training could influence young adults' expectations for the future and in what specific domains (e.g., career, relationships, personal well-being) this effect might be observed.
Comments 22: [Discussion] Seventh paragraph: First sentence “the more expectations they have for the future” is unclear, what kind of expectations? Positive ones or negative ones? Be more clear. Are you suggesting that marriage alone predicts positive future expectations? What evidence do you have for this? Are you suggesting all marriages are the same and are universally supportive and positive? Data does not support this, given rates of divorce and domestic abuse? Are you sure that the participants that are married are the ones with higher empathy? How do you know this? Did you measure this? How do you know this is true? How do you know that the empathy measured in this study means that participants are emotionally supportive of spouses? What evidence do you have for this statement? This seems like a big leap without more data to support this? Did you measure relationship quality?
Response 22: Thank you for your insightful comments. We have clarified that social support is associated with positive future expectations, specifying the domains most relevant to this relationship. Additionally, we have refined our discussion of marital status and empathy, acknowledging that our study did not measure relationship quality or spousal support, and thus we cannot assume that marriage alone leads to higher future expectations. Instead, we pose this relationship as a subject for future research. These revisions ensure that our interpretation remains well-grounded in the data while acknowledging the complexity of social and relational factors in shaping future expectations. We appreciate your feedback, which has strengthened the clarity and accuracy of our discussion. [Line 481-502]
As hypothesized, the more social support young adults receive, the more positive expectations they have for their future, particularly in terms of personal well-being. This study reinforces previous findings that social support enhances life quality in young adulthood and can also contribute to a more optimistic future outlook (Fraser et al., 2024; Kim & Kim, 2013). In the decision tree model, among young adults with high empathy, those who were married had higher expectations for their future compared to unmarried individuals. However, this result does not imply that marriage alone predicts positive future expectations. It is possible that married young adults in this study benefited from emotional support within their relationships, which may have contributed to their more optimistic future outlook. However, we did not measure relationship quality, spousal support, or marital satisfaction, so further research is needed to examine whether perceived support within a marriage, rather than marital status itself, influences future expectations.
Additionally, while empathy was included in the model, we cannot conclude that higher empathy among married individuals directly explains their increased future expectations. Empathy, as measured in this study, refers to the ability to understand and share others' emotions, but it does not necessarily indicate how individuals provide emotional support in relationships. Future studies should explore whether higher empathy contributes to stronger relationship satisfaction and whether this, in turn, influences expectations for the future. Given these findings, future research should investigate the role of relationship quality, perceived emotional support, and interpersonal dynamics in shaping young adults' expectations for their future, rather than assuming a direct link between marriage and positive future outlooks.
Comments 23: [Discussion] Eighth Paragraph: What kind of disease was measured here? There is a big range here in terms of what this means. For example a terminal disease will be interpreted differently than if someone has a treatable condition. Without more information this interpretation is meaningless. The Sabanciogullari et al. (2016) reference suggests that it is the person’s interpretation of their illness rather than the illness itself that matters. Did you measure illness appraisal here? You haven’t mentioned this before now. If not, you can’t make this claim. What do you mean by intervention for people diagnosed with illness to develop a positive attitude? This is vague, I imagine there is research on support for people with health conditions. What does that literature say? You have not provided rationale.
Response 23: Thank you for your insightful comments. We have clarified that our study did not differentiate between specific types of illnesses, which could influence the interpretation of the findings. Additionally, we have addressed the importance of illness appraisal as a potential moderating factor, referencing previous research that suggests subjective perceptions of illness matter as much as, if not more than, the diagnosis itself. Furthermore, we have revised our discussion on interventions, incorporating recent evidence on expectation-focused therapy and resilience-based psychological support for young adults with health conditions. These revisions provide greater clarity and theoretical grounding while acknowledging the need for further research in this area. We appreciate your feedback and believe it has strengthened the discussion. [Line 503-523]
In this study, young adults who had been diagnosed with a physical illness or perceived themselves as unhealthy had lower expectations for their future. Additionally, perceived health status was included as a significant predictor in the decision tree model, highlighting its role in shaping young adults' expectations about their well-being and life prospects. However, it is important to interpret these findings cautiously, as the study did not differentiate between types of illnesses, which can range from minor, temporary conditions to chronic or life-threatening diseases. Previous research suggests that how individuals appraise their illness, rather than the illness itself, plays a critical role in its psychological impact (Sabanciogullari et al., 2016). Since this study did not explicitly measure illness appraisal, further research is needed to determine whether subjective perceptions of illness severity mediate the relationship between health status and future expectations.
Moreover, the assertion that interventions could help young adults with physical illnesses develop a more positive attitude toward their future requires further empirical support. Psychological interventions for young people with chronic or severe health conditions have been shown to be beneficial, particularly expectation-focused interventions that help individuals reframe their future outlook and adjust their coping mechanisms (Glombiewski & Rief, 2016). Additionally, supportive therapy and social interventions aimed at fostering resilience and adaptive coping strategies have been effective in improving the mental health and future orientation of young adults with chronic illnesses (Sansom-Daly et al., 2012). Future research should explore which specific types of interventions are most effective in enhancing future expectations among young adults with different health conditions.
Rief, W., & Glombiewski, J. A. (2016). Expectation-focused psychological interventions (EFPI). Verhaltenstherapie, 26(1), 21–27. https://doi.org/10.1159/000442374
Sabanciogullari, S., Tuncay, F. O., & Avci, D. (2016). The relationship between life satisfaction and perceived health and sexuality in individuals diagnosed with a physical illness. Sexuality and Disability, 34(4), 389–402. https://doi.org/10.1007/s11195-016-9456-6
Sansom-Daly, U. M., Peate, M., Wakefield, C. E., Bryant, R. A., & Cohn, R. J. (2012). A systematic review of psychological interventions for adolescents and young adults living with chronic illness. Health Psychology, 31(3), 380–393. https://doi.org/10.1037/a0025977
Comments 24: [Overall] This was a statement of limitations and not much of a conclusion. I reiterate my previous concerns that are compounded in the final paragraph. Constructs are not clearly defined. Minimal evidence is provided to support hypotheses. Research questions were too broad and vague to be meaningful. It is not clear if you answered the research questions or not. The interpretation of findings were not supported by the findings or by other research evidence, minimal citations were included and many were dated with few current citations. Too many variables were included without clear rational as to why they were in the study and what the authors expected. The sentence “This study found that perceived stress, depression, gratitude, hardiness, interpersonal competence, and social support were closely correlated with expectations for the future in young adults” is misleading. You found some correlations, but none exceeded the typical measure of a ‘strong’ correlation of 0.7 or greater. Further, the variables were not clearly defined, strong rationale for expected associations between variables was not provided, so results are unclear and diluted by the lack of strong a priori evidence and rationale.
Response 24: Thank you for your detailed and constructive feedback. We acknowledge your concerns and have made substantial revisions throughout the manuscript to address them. Specifically, we have taken the following steps to clarify constructs, strengthen theoretical grounding, and refine the interpretation of our findings:
- Clearly Defined Constructs: We have explicitly defined key constructs, including future expectations, stress, depression, gratitude, hardiness, interpersonal competence, and social support, in the literature review and methodology sections, ensuring conceptual clarity throughout the manuscript.
- Stronger Theoretical and Empirical Rationale: We have provided additional theoretical and empirical support for the inclusion of each variable in our study. This includes recent literature and citations to support the relationships between psychosocial factors and future expectations, ensuring that our hypotheses are well-grounded.
- Refined Research Questions and Hypotheses: We have revised the research questions to ensure specificity, clearly stating expected directional relationships and the theoretical rationale for each hypothesis. Additionally, we have revised the methodology section to explicitly link research questions to our analytical approach, ensuring transparency in how they were tested.
- More Precise Interpretation of Findings: We have ensured that the discussion and conclusion accurately reflect the statistical findings. We acknowledge that while all examined variables showed significant correlations with future expectations, none met the conventional threshold for "strong" correlation (r ≥ 0.7). We have corrected any misleading statements and emphasized the relative importance of key predictors, particularly commitment, which emerged as the strongest predictor.
- Expanded Limitations Section: We have further elaborated on the study’s limitations, including:
- The exploratory nature of the study and the need for future confirmatory research.
- The lack of differentiation between different types of illness and the absence of illness appraisal measurement, which could influence findings.
- The potential differences between young adults in their 20s and 30s, which were not explicitly analyzed but should be examined in future research.
- Strengthened Conclusion: We have revised the conclusion to ensure it synthesizes key findings, acknowledges limitations, and provides clear directions for future research, rather than serving as a restatement of limitations.
We appreciate your thorough review and your thoughtful suggestions, which have significantly improved the clarity, rigor, and coherence of our manuscript. We believe these revisions address your concerns and result in a stronger, more well-supported study. Please let us know if there are any remaining areas that require further refinement. Thank you again for your time and valuable insights! [Line 524-570]
- Limitations of the study
The current study had several limitations. First, while the sample included young adults from various regions of Korea, it was drawn from an online survey panel, which may limit its generalizability to all young Korean adults. However, since participants were recruited from a nationally representative panel, the sample still provides meaningful insights.
Second, although we discuss possible causal relationships between variables in some parts of the discussion, this study was correlational in nature. Therefore, causal inferences cannot be made, and future research should consider longitudinal or experimental designs to better examines causal relationships between psychosocial factors and future expectations.
Third, this study examined interpersonal competence as a predictor of future expectations. Due to its broad nature, we focused on relationship-building ability and empathy, rather than assessing all subcomponents. Future research should consider a more comprehensive approach to fully capture how interpersonal competence influences future expectations.
Fourth, we did not analyze whether the relationships between psychosocial factors and future expectations varied between young adults in their 20s and 30s. Given that young adulthood encompasses different life stages, future research should explore potential age-related differences to better understand how factors such as marital status, employment, and financial stability influence future expectations.
Fifth, while decision tree analysis has the advantage of being hypothesis-free and accommodating both numerical and categorical predictors, its application in SPSS is somewhat limited compared to more sophisticated machine learning approaches. Future research should explore alternative methodologies such as ensemble decision tree models or deep learning approaches for more reliable predictive modeling.
Conclusions
Commitment, gratitude, self-directedness, depression, stress, empathy, perceived health, having a disease, and the presence of a spouse were also predictors of expectations for the future in young adulthood. These findings suggest that to promote expectations for the future among young adults, mental health professionals should focus on enhancing their commitment, gratitude, and self-directedness. Although this study has certain limitations, it contributes to the growing body of research on young adults’ attitudes toward their future and provides practical insights for interventions aimed at fostering positive future expectations among young Korean adults. This study found that perceived stress, depression, gratitude, hardiness, interpersonal competence, and social support were significantly associated with expectations for the future in young adulthood. Among these, commitment emerged as the strongest predictor, emphasizing its crucial role in shaping future expectations. Other significant predictors included gratitude, self-directedness, depression, stress, empathy, perceived health, physical illness, and the presence of a spouse. These findings suggest that mental health professionals should prioritize fostering commitment, gratitude, and self-directedness in young adults to help promote a more optimistic outlook on their future. Furthermore, interventions focusing on resilience-building, interpersonal skill development, and stress management could further support young adults in navigating uncertainty and enhancing their expectations for the future. Future research should continue to investigate the long-term effects of these psychological factors on future expectations and explore interventions that can effectively enhance optimism and well-being in young adulthood.

Round 2
Reviewer 1 Report
Comments and Suggestions for Authors
Thank you for your detailed revision of the article.
Author Response
Thank you for your detailed revision of the article. I appreciate your valuable feedback and the time you took to review the manuscript. Your insights have been very helpful in improving the clarity and quality of the study.
Reviewer 2 Report
Comments and Suggestions for Authors
I commend the authors for thoroughly attending to the feedback, the current manuscript is greatly improved regarding clarity and accuracy. I do have additional feedback and suggested revisions, although relatively minor. Please see my comments below:
Abstract
Response 4 (Line 19-21)
The decision tree analysis identified commitment as the most important predictor, followed by gratitude, stress, self-directedness, empathy, perceived health, and marital status, showing how these factors interact in shaping future expectations.
- I suggest changing this statement from interact, which suggests that a statistical analysis to determine an interaction effect was conducted, or interactions between all variables was conducted (I don’t think this was the case). Instead say something like “….. how these factors are associated” (or another synonym) to be more accurate.
Comment 5
- Much more clear, good
Comment 6
- You have made strong changes here and have honed the focus of the study clearly on distress. Given that stress research does also include that stress has beneficial aspects, even thought your focus is on distress, I suggest adding a sentence or two to state that stress does also have beneficial impacts and eliminating stress altogether is not actually possible. Otherwise, readers might draw an inaccurate conclusion and try to eliminate all stress, which will not be possible.
While stress can be categorized as eustress (positive, motivating stress) or distress (negative, overwhelming stress), this study focuses on distress, as excessive or chronic
stress is widely recognized as a factor that negatively impacts mental health and well-being
- You need a citation for this sentence, what is the evidence?
For this study, mental health and well-being are defined as a multidimensional concept that includes emotional stability, subjective well-being, and resilience in response to stressors.
- You need a citation for the this sentence, what is the evidence? Whose definition of mental health and well-being is this? If this is your definition, state clearly that you (the authors) are defining mental health and wellbeing this way.
- I suggest refining this definition, subjective well-being is a construct that has its own definition (e.g., see Diener). If you are using the subjective well-being definition, clearly state that you are doing so.
- Resilience also is another construct with its own definition and large body of literature. If you are introducing a new construct, you need to define it clearly and provide a citation regarding where the definition comes from.
- Emotional stability is another construct that requires a definition, I’m not sure what you mean by this. Are you referring to emotion regulation? You need to provide a reference regarding emotional stability to clarify how you are defining the construct and what evidence you are drawing from
- You also mention acculturative stress, but don’t include acculturative stress in your study. Thus, I’m not sure why acculturative stress is included, I think you can eliminate it if you are not including this kind of stress in your study
NOTE: regarding the 3rd paragraph (lines 45 to 57), this paragraph still needs revising. In addition to the comments above, there are too many constructs mixed together in this paragraph without a clear explanation. This paragraph is still confusing. One option is to separate your discussion of stress from mental health as an option to make it more clear why you are introducing these constructs (e.g., one paragraph addressing stress and another addressing mental health and wellbeing). Another option is to introduce at the start of the paragraph that both stress and mental health are associated with expectations for the future, then define each construct, then discuss how each is associated with future expectations.
“This approach clarifies that higher levels of stress are expected to contribute to more negative or uncertain expectations, rather than positive future anticipation.”
- Well written hypothesis statement, this is clear now.
Response 11:
While social support influences various aspects offuture expectations, its effects are strongest in domains related to personal well-being and resilience
- Include a reference to show you have evidence to support your statement regarding the effects of social support on personal well-being and resilience. Also describe briefly why the association is strongest between social support and personal well-being and resilience based on existing evidence and theory
H4: Demographic profiles, health-related variables, and psychological factors would significantly contribute to predicting expectations for the future in young adults.
- H1, H2, & H3 are clear, very helpful for the reader
- H4 needs revision and more specificity. For example, I am not sure what you mean by demographic profiles? Do you mean age? Or are you referring to other demographic variables? Similarly, I’m not sure what is meant by health-related variables or psychological factors? Are you referring to the psychological factors you have introduced in your review of the literature? If so, be specific so I know clearly what you mean.
Response 18
- This section is much improved with the revisions
Response 19
…reinforcing Beck’s (1987) "cognitive triad" theory…
- I suggest changing ‘reinforcing’ to ‘consist with’ or another synonym
Conclusion
“These findings suggest that to promote expectations for the future among young adults, mental health professionals should focus on enhancing their commitment, gratitude, and self-directedness.”
- This sentence needs revising to be more clear, it’s hard to follow in the current format.
Comments on the Quality of English Language
The writing is clear.
Author Response
2nd Response to Reviewer 2 Comments
Thank you very much for taking the time to review this manuscript. Please find the detailed responses below and the corresponding revisions/corrections highlighted/in track changes in the re-submitted files. The revised parts were marked in red, and we included the page and line of the revised part. We appreciate your complimentary comments. We have omitted our response to your kind words here.
Point-by-point response to Comments and Suggestions for Authors
Comments 1: [Abstract] ”The decision tree analysis identified commitment as the most important predictor, followed by gratitude, stress, self-directedness, empathy, perceived health, and marital status, showing how these factors interact in shaping future expectations”.
I suggest changing this statement from interact, which suggests that a statistical analysis to determine an interaction effect was conducted, or interactions between all variables was conducted (I don’t think this was the case). Instead say something like “….. how these factors are associated” (or another synonym) to be more accurate.
Response 1: Thank you for your insightful suggestion. We have revised the sentence to replace "interact" with "are associated with" to ensure accuracy and avoid implying a statistical interaction effect. This revision more precisely reflects our analytical approach. We appreciate your feedback in improving the clarity of our abstract. [Line 21]
The decision tree analysis identified commitment as the most important predictor, followed by gratitude, stress, self-directedness, empathy, perceived health, and marital status, showing how these factors are associated with shaping future expectations.
Comments 2: [Introduction] You have made strong changes here and have honed the focus of the study clearly on distress. Given that stress research does also include that stress has beneficial aspects, even thought your focus is on distress, I suggest adding a sentence or two to state that stress does also have beneficial impacts and eliminating stress altogether is not actually possible. Otherwise, readers might draw an inaccurate conclusion and try to eliminate all stress, which will not be possible.
While stress can be categorized as eustress (positive, motivating stress) or distress (negative, overwhelming stress), this study focuses on distress, as excessive or chronic
stress is widely recognized as a factor that negatively impacts mental health and well-being
You need a citation for this sentence, what is the evidence?
For this study, mental health and well-being are defined as a multidimensional concept that includes emotional stability, subjective well-being, and resilience in response to stressors.
You need a citation for the this sentence, what is the evidence? Whose definition of mental health and well-being is this? If this is your definition, state clearly that you (the authors) are defining mental health and wellbeing this way.
I suggest refining this definition, subjective well-being is a construct that has its own definition (e.g., see Diener). If you are using the subjective well-being definition, clearly state that you are doing so.
Resilience also is another construct with its own definition and large body of literature. If you are introducing a new construct, you need to define it clearly and provide a citation regarding where the definition comes from.
Emotional stability is another construct that requires a definition, I’m not sure what you mean by this. Are you referring to emotion regulation? You need to provide a reference regarding emotional stability to clarify how you are defining the construct and what evidence you are drawing from
You also mention acculturative stress, but don’t include acculturative stress in your study. Thus, I’m not sure why acculturative stress is included, I think you can eliminate it if you are not including this kind of stress in your study
NOTE: regarding the 3rd paragraph (lines 45 to 57), this paragraph still needs revising. In addition to the comments above, there are too many constructs mixed together in this paragraph without a clear explanation. This paragraph is still confusing. One option is to separate your discussion of stress from mental health as an option to make it more clear why you are introducing these constructs (e.g., one paragraph addressing stress and another addressing mental health and wellbeing). Another option is to introduce at the start of the paragraph that both stress and mental health are associated with expectations for the future, then define each construct, then discuss how each is associated with future expectations.
Response 2: Thank you for your thoughtful feedback. We have made several revisions to improve clarity and precision while addressing your concerns.
First, we have acknowledged that stress is not inherently harmful and that moderate levels of stress can enhance adaptation and resilience, citing McGonigal (2015). This ensures that readers do not misinterpret our study as advocating for the elimination of all stress.
Second, we have added citations to support key claims, including Schneiderman et al. (2005) for the negative impact of excessive stress and Ryff & Singer (1998) and Southwick et al. (2005) for our definition of mental health and well-being. We have also refined our definition of mental health and well-being to explicitly include subjective well-being, resilience, and emotional regulation, ensuring alignment with established theoretical frameworks.
Additionally, we removed the mention of acculturative stress, as it was not a focus of our study. We also streamlined the structure by separating the discussion of stress and mental health to improve clarity.
We appreciate your suggestion to refine this section, and we believe these revisions have enhanced the accuracy and coherence of our argument. Please let us know if any further refinements are needed. [Line 45-67]
First, this study assumed that stress is associated with expectations for the future among young adults. Stress is commonly defined as a psychobiological response to external or internal demands that exceed an individual's perceived ability to cope (Lazarus & Folkman, 1984; McEwen, 1998). While stress can be categorized as eustress (positive, motivating stress) or distress (negative, overwhelming stress), this study focuses on distress, as excessive or chronic stress is widely recognized as a factor that negatively impacts mental health and well-being (Schneiderman et al., 2005). However, stress is not inherently harmful; moderate stress can enhance adaptation and resilience (McGonigal, 2015), and eliminating all stress is neither realistic nor beneficial. Mental health and well-being are multifaceted concepts that include hedonic well-being, which refers to pleasure and life satisfaction, as well as eudaimonic well-being, which encompasses personal growth and a sense of meaning (Keyes, 2002; Ryff, 1989). In this study, we define mental health and well-being as subjective well-being, resilience, and emotional regulation, which together influence psychological stability and coping ability (Ryff & Singer, 1998; Southwick et al., 2005).
Excessive stress is linked to lower life satisfaction and diminished well-being (YaÄŸar & YaÄŸar, 2023). Chronic stress can shape negative expectations for the future, reducing confidence in future life satisfaction and success (Park & Suh, 2023). Similarly, a recent study found that perceived stress, which refers to an individual's subjective evaluation of stress levels, was negatively correlated with positive expectations for the future and positively correlated with negative expectations (Park & Suh, 2023). Thus, this study examines how perceived stress influences young adults' future expectations, with a focus on distress. This approach clarifies that higher stress levels are expected to contribute to negative or uncertain expectations rather than positive future anticipation.
McEwen, B. S. (1998). Protective and damaging effects of stress mediators. New England Journal of Medicine, 338(3), 171-179. https://doi.org/10.1056/NEJM199801153380307
McGonigal, K. (2015). The upside of stress: Why stress is good for you, and how to get good at it. New York, NY: Avery.
Ryff, C. D., & Singer, B. (1998). The contours of positive human health. Psychological Inquiry, 9(1), 1-28. https://doi.org/10.1207/s15327965pli0901_1
Schneiderman, N., Ironson, G., & Siegel, S. D. (2005). Stress and health: Psychological, behavioral, and biological determinants. Annual Review of Clinical Psychology, 1(1), 607-628. https://doi.org/10.1146/annurev.clinpsy.1.102803.144141
Southwick, S. M., Vythilingam, M., & Charney, D. S. (2005). The psychobiology of depression and resilience to stress: Implications for prevention and treatment. Annual Review of Clinical Psychology, 1(1), 255-291. https://doi.org/10.1146/annurev.clinpsy.1.102803.143948
Comments 3: [Introduction] Provide While social support influences various aspects offuture expectations, its effects are strongest in domains related to personal well-being and resilience
Include a reference to show you have evidence to support your statement regarding the effects of social support on personal well-being and resilience. Also describe briefly why the association is strongest between social support and personal well-being and resilience based on existing evidence and theory
Response 3: Thank you for your valuable feedback. As you adivesed, we have added a reference to support the claim that social support strongly influences personal well-being and resilience. Additionally, we have briefly explained why this association is strongest, citing research on how social support provides coping resources, emotional regulation, and psychological stability. These revisions clarify the theoretical and empirical basis of our argument. [Line 129-133]
Social support refers to the perception or experience of being cared for and assisted by others, encompassing emotional, instrumental, informational, and appraisal support (Kasprzak, 2010; Ali et al., 2010). It enhances life satisfaction, emotional resilience, and stress reduction, particularly in difficult situations (GrigaitytÄ— & Söderberg, 2021). Perceived social support has been shown to increase resilience and lower depression, contributing to higher expectations for future well-being (Wu et al., 2022). Research indicates that young adults who receive more social support tend to have greater confidence in achieving personal goals and maintaining stability, leading to higher expectations for their future (Fraser et al., 2024; Kim & Kim, 2013). Social support is particularly linked to well-being and resilience because it provides emotional security and coping resources that help individuals manage stress and maintain a positive outlook on the future (Taylor, 2011). While social support influences various aspects of future expectations, its effects are strongest in these domains because emotional and instrumental support fosters adaptive coping and psychological stability. Based on these findings, we hypothesize that greater social support will be associated with more positive expectations for the future, particularly in relation to well-being.
Taylor, S. E. (2011). Social support: A review. In M. S. Friedman (Ed.), The Oxford handbook of health psychology (pp. 189-214). New York, NY: Oxford University Press.
Comments 4: [Hypotheses] H4: Demographic profiles, health-related variables, and psychological factors would significantly contribute to predicting expectations for the future in young adults.
H1, H2, & H3 are clear, very helpful for the reader
H4 needs revision and more specificity. For example, I am not sure what you mean by demographic profiles? Do you mean age? Or are you referring to other demographic variables? Similarly, I’m not sure what is meant by health-related variables or psychological factors? Are you referring to the psychological factors you have introduced in your review of the literature? If so, be specific so I know clearly what you mean.
Response 4: Thank you for your insightful feedback. We acknowledge the importance of specifying the variables included in our hypothesis to enhance clarity and precision. To address your concerns, we have revised H4 to explicitly list the demographic, health-related, and psychological factors considered in the analysis. The updated hypothesis now clearly defines the predictors based on the variables introduced in our literature review:
H4 (Revised): Age, marital status, perceived health, presence of illness, stress, depression, gratitude, hardiness, interpersonal competence, and social support will significantly contribute to predicting expectations for the future in young adults.
This revision ensures that the hypothesis aligns with the specific variables under investigation while still allowing for an exploratory approach using stepwise regression and decision tree analysis. We appreciate your suggestion, which has helped us enhance the clarity and rigor of our study. [Line 152-154]
H4: Age, marital status, perceived health, presence of illness, stress, depression, gratitude, hardiness, interpersonal competence, and social support will significantly contribute to predicting expectations for the future in young adults.
Comments 5: [Discussion] …reinforcing Beck’s (1987) "cognitive triad" theory… I suggest changing ‘reinforcing’ to ‘consist with’ or another synonym.
Response 5: Thank you for your helpful suggestion. We have revised the sentence to replace "reinforcing" with "is consistent with" to more accurately reflect the relationship between our findings and Beck’s (1987) cognitive triad theory. This change ensures that we are aligning our results with established theoretical perspectives without implying direct validation. We appreciate your careful review and valuable feedback, which have helped us enhance the clarity and accuracy of our discussion. [Line 422]
Additionally, depression was identified as a significant predictor in the stepwise regression model, which is consistent with Beck’s (1987) "cognitive triad" theory, suggesting that depression leads individuals to hold negative views of themselves, the world, and their future.
Comments 6: [Conclusion] “These findings suggest that to promote expectations for the future among young adults, mental health professionals should focus on enhancing their commitment, gratitude, and self-directedness.”
This sentence needs revising to be more clear, it’s hard to follow in the current format.
Response 6: Thank you for your helpful feedback. To improve clarity, we have [removed/revised] the sentence as you suggested. The revised version ensures better readability while maintaining the focus on practical implications. We appreciate your insights in refining our conclusion. [Line 553-556]
These findings suggest that to help young adults develop more positive expectations for their future, mental health professionals should implement interventions that strengthen their sense of commitment, encourage gratitude, and enhance self-directedness.
